# The SSVEP tracks attention, not consciousness, during perceptual filling-in

**Matthew J Davidson[1,2]\*, Will Mithen[1], Hinze Hogendoorn[3], Jeroen JA van Boxtel[4†], Naotsugu Tsuchiya[1,5,6,7†]**

[1]School of Psychological Sciences, Faculty of Medicine, Nursing and Health Science, Monash University, Melbourne, Australia; [2]Department of Experimental Psychology, Faculty of Medicine, University of Oxford, Oxford, United Kingdom; [3]Melbourne School of Psychological Sciences, University of Melbourne, Melbourne, Australia; [4]Discipline of Psychology, Faculty of Health, University of Canberra, Canberra, Australia; [5]Turner Institute for Brain and Mental Health, Faculty of Medicine, Nursing and Health Science, Monash University, Melbourne, Australia; [6]Center for Information and Neural Networks (CiNet), National Institute of Information and Communications Technology (NICT), Suita, Japan; [7]Advanced Telecommunications Research Computational Neuroscience Laboratories, 2-2-2 Hikaridai, Seika-cho, Soraku-gun, Kyoto, Japan

**Abstract** Research on the neural basis of conscious perception has almost exclusively shown that becoming aware of a stimulus leads to increased neural responses. By designing a novel form of perceptual filling-in (PFI) overlaid with a dynamic texture display, we frequency-tagged multiple disappearing targets as well as their surroundings. We show that in a PFI paradigm, the disappearance of a stimulus and subjective invisibility is associated with increases in neural activity, as measured with steady-state visually evoked potentials (SSVEPs), in electroencephalography (EEG). We also find that this increase correlates with alpha-band activity, a well-established neural measure of attention. These findings cast doubt on the direct relationship previously reported between the strength of neural activity and conscious perception, at least when measured with current tools, such as the SSVEP. Instead, we conclude that SSVEP strength more closely measures changes in attention.

**\*For correspondence:**
mjd070@gmail.com

[†]These authors contributed equally to this work

**Competing interests:** The authors declare that no competing interests exist.

## Introduction

A central question in cognitive neuroscience is asking what the neural correlates or constituents of consciousness may be (*Klink et al., 2015*; *Koch et al., 2016*; *Mashour et al., 2020*; *Miller, 2007*). Recent empirical research suggests that various psychological processes, such as cognitive access, expectation, and the act of report, may confound our interpretation of these correlates (*Aru et al., 2012*; *Frässle et al., 2014*; *Tse et al., 2005*; *Tsuchiya et al., 2015*; *van Boxtel and Tsuchiya, 2015*; *van Boxtel et al., 2010a*). Among them, one of the most hotly debated is task-relevant attention and its relation to visual phenomenology (*Block, 2011*; *Cohen et al., 2016*; *Koch and Tsuchiya, 2007*; *Kouider et al., 2010*; *Lamme, 2006*; *Lau and Rosenthal, 2011*; *Posner, 2012*; *Tsuchiya and Koch, 2016*). Are attention and visual experience always tightly intertwined, or can they be disentangled? Despite accumulating behavioral evidence in favor of the latter, it remains unclear whether the neural mechanisms of attention and visual awareness are distinct. Notwithstanding new methodological developments (*Maier and Tsuchiya, 2020*; *Smout and Mattingley, 2018*; *Watanabe et al., 2011*; *Webb et al., 2016*), there is still an unmet empirical need for experimental paradigms which

can address this question (*Pitts et al., 2018*; *Tallon-Baudry, 2011*; *van Boxtel, 2017*; *van Boxtel and Tsuchiya, 2015*; *van Boxtel et al., 2010a*).

Perceptual filling-in (PFI) is one candidate paradigm that can potentially dissociate attention from consciousness. During PFI, a distinct region of the visual periphery disappears and becomes interpolated by the surrounding image background. PFI is a relatively underexplored class of multistable phenomena (*Davidson et al., 2020*; *Kim and Blake, 2005*; *Sterzer et al., 2009*), but it has distinct advantages over its predecessors (*Komatsu, 2006*; *Weil and Rees, 2011*). For example, compared to more widely studied forms of multistable stimuli that require inter-ocular suppression, PFI occurs regularly in normal vision (*Anstis and Greenlee, 2014*; *De Weerd, 2006*; *Durgin et al., 1995*), as in the visual areas that are occluded by the blindspot, and the blood vessels in the retina (*Komatsu, 2006*).

In the behavioral and neuronal study of consciousness and attention, PFI has an unusual and potentially very revealing property: directing attention toward targets in the visual periphery *facilitates* perceptual disappearance (*De Weerd et al., 2006*; *Lou, 1999*). This perceptual property is rare, but not unique to PFI. It has been observed in research on afterimages (*Lou, 2001*; *van Boxtel et al., 2010b*), motion aftereffects (*Murd and Bachmann, 2011*), and motion induced-blindness (*Geng et al., 2007*; *Schölvinck and Rees, 2009*). Outside the literature in perception, attention can also harm performance in tasks that involve memory (*Voss and Paller, 2009*) and skilled motor performance (*Beilock et al., 2002*; *Logan and Crump, 2009*). However, PFI, together with motion induced-blindness (*Geng et al., 2007*; *Schölvinck and Rees, 2009*), is arguably the most versatile tool amongst these phenomena, particularly as the use of multiple disappearing targets can enable a graded measure of visual phenomenology to be recorded (*Davidson et al., 2020*). Since PFI can dissociate attention and consciousness at the perceptual level, in strictly behavioral paradigms, it has been proposed as a highly promising paradigm to dissociate the neural correlates of attention and consciousness as well (*Tsuchiya and Koch, 2016*). Here, we demonstrate that PFI indeed dissociates neural measures which normally positively correlate with both attention to, and the conscious perception of, visual stimuli.

As neural measures of stimulus disappearance and reappearance during PFI, we focus on steady-state visually evoked potentials (SSVEPs). SSVEPs are periodic large-scale cortical responses to rhythmic visual stimuli, 'frequency-tagging' a neural population specific to each flickering stimulus for subsequent analysis (*Tononi et al., 1998*). SSVEPs have a high signal-to-noise ratio (SNR), and as a result have been widely adopted in the basic visual, cognitive, and clinical neurosciences (*Norcia et al., 2015*; *Vialatte et al., 2010*).

Importantly, SSVEPs have been extensively used as the neural correlates of visual consciousness, predominantly when combined with binocular rivalry (*Brown and Norcia, 1997*; *Cobb et al., 1967*; *Katyal et al., 2016*; *Lansing, 1964*; *Tononi et al., 1998*; *Zhang et al., 2011*). During binocular rivalry, dissimilar monocular images presented to each eye compete for perceptual dominance, with conscious awareness typically fluctuating spontaneously between each image (*Alais, 2012*; *Alais and Blake, 2005*; *Levelt, 1965*; *Maier et al., 2012*). When two flickering stimuli compete during rivalry, the relative strength of frequency-tagged activity positively correlates with behavioral reports of the dominant image (*Bock et al., 2019*; *Brown and Norcia, 1997*; *Cosmelli et al., 2004*; *Jamison et al., 2015*; *Kamphuisen et al., 2008*; *Katyal et al., 2016*; *Lansing, 1964*; *Lawwill and Biersdorf, 1968*; *Roy et al., 2017*; *Srinivasan et al., 1999*; *Sutoyo and Srinivasan, 2009*; *Tononi et al., 1998*; *Zhang et al., 2011*).

In a largely separate and parallel body of literature, SSVEPs have also been employed as a measure of selective attention, which increases the strength of SSVEP responses (*Morgan et al., 1996*; *Müller et al., 2006*, *Müller et al., 1998*; *Müller and Hillyard, 2000*; *Müller and Hübner, 2002*; *Walter et al., 2012*). Notably, an increase in SSVEP strength when stimuli are attended also occurs when flickering items are presented simultaneously in close proximity (*Andersen et al., 2008*; *Müller et al., 2006*; *Wang et al., 2007*).

In their recent study, *Smout and Mattingley, 2018*, further demonstrated the potential of the SSVEP to dissociate the neural effects of attention from consciousness. Specifically, although their participants were unaware of an SSVEP-inducing stimulus embedded in noise, allocating spatial attention to the side of the invisible stimulus enhanced the strength of the SSVEP (*Smout and Mattingley, 2018*). This result extended previous behavioral and neural studies showing that processing

related to an invisible stimulus can be enhanced by attention (*Cohen et al., 2016*; *Koch and Tsuchiya, 2007*; *Tsuchiya and Koch, 2016*; *van Boxtel et al., 2010a*).

Despite these advantages, SSVEPs have never been applied to experimental paradigms where the effects of conscious visibility and focus of attention on neuronal activity are expected to diverge. Here, we employed a PFI paradigm to directly induce such a situation, as the probability of a stimulus disappearing (i.e. being filled-in) increases, rather than decreases, with attention (*De Weerd et al., 2006*; *Lou, 1999*). We employed a multi-target, multi-response paradigm (*Davidson et al., 2020*), in which all disappearing targets were matched in color, form, and temporal flicker, and tasked participants to covertly monitor all targets for instances of PFI. Thus, the neuronal effects of consciousness and attention are expected to be in opposition in this paradigm, if attention to the targets facilitates their disappearance.

Importantly, we developed a novel method that allowed us to capture neural markers of attention and/or consciousness during PFI, that separately frequency-tagged targets and their visual surrounds. We achieved this by building upon previous work that has facilitated PFI by making the surrounding regions dynamic, such as when updated with visual noise (*Davidson et al., 2020*; *De Weerd et al., 1998*; *De Weerd et al., 1995*; *Ramachandran and Gregory, 1991*; *Spillmann and Kurtenbach, 1992*; *Weil et al., 2007*; *Weil et al., 2008*). Our critical manipulations were to calibrate the dynamic noise texture to evoke an SSVEP and to extend this texture over our enlarged target regions. While target size normally impedes perceptual disappearances during PFI (*De Weerd et al., 1998*), we included a pre-experiment calibration session to ensure frequent occurrences of large parafoveal PFI for each participant (*Figure 1*).

With this innovation, we were able to dynamically update our visual texture within a target boundary at 15 Hz (F1), and the target surrounds at a rate of 20 Hz (F2; Movie 1, https://osf.io/kdrz3/, *Figure 1*). This approach, which we call *dynamic texture tagging*, allowed us to monitor changes to the time-course of target and surround specific neural activity at different frequencies, as well as their interaction at intermodulation frequencies (for a review on intermodulation components in cognitive neuroscience, see *Gordon et al., 2019*). We expected target and surround-specific neural signals to anti-correlate if the SSVEP signal is related to visibility, as the visibility of flickering target and surround stimuli are inversely correlated during filling-in. If the SSVEP is related to attention, however, we expected these signals to correlate positively, because attending to the spatial location of a filling-in target would enhance the strength of both surround and (invisible) target SSVEPs around the time of disappearance.

## Results

### Target-SNR strength dissociates from visual phenomenology during PFI

Our participants ($N$ = 16) reported on the subjective invisibility of four simultaneously presented parafoveal targets, located in separate visual quadrants (*Figure 1*). We first confirmed in our behavioral analysis that targets disappeared together more frequently than in isolation (see Supplementary results, *Figure 2—figure supplement 1*). Consistent with our prior work, we also replicate that the duration of PFI increases with an increasing number of invisible targets, strongly suggesting a facilitatory grouping mechanism across visual hemifields and target locations (*Davidson et al., 2020*).

Next, we analyzed the time-course of target (f1) and surround (f2) SSVEP-SNR during PFI, aligned to button presses at target disappearance, and button releases at reappearance. We found that the strength of target-specific neural responses increased prior to the subjective report of a target becoming invisible. *Figure 2a* compares the target-related SNR when aligned to target disappearances (o) versus reappearances (-). The magnitudes of each time course differ significantly from −0.74 s before, through to 1.53 s after subjective report ($p_{cluster}$ <0.001). During this period, target-SNR is negatively correlated with the contents of conscious perception during PFI; when a target disappears, target-SNR increases. This is consistent with our prior expectation that SSVEP strength during PFI may diverge from a measure of visual consciousness. As discussed below (and shown in *Figure 2b*), a similar divergence does not occur for physical disappearances.

Next, we sorted all instances of PFI by the number of buttons that were pressed simultaneously, measuring the 'amount of PFI', which we defined as the duration-weighted number of invisible targets (see Materials and methods). We found that increases in the amount of PFI (i.e., decreased

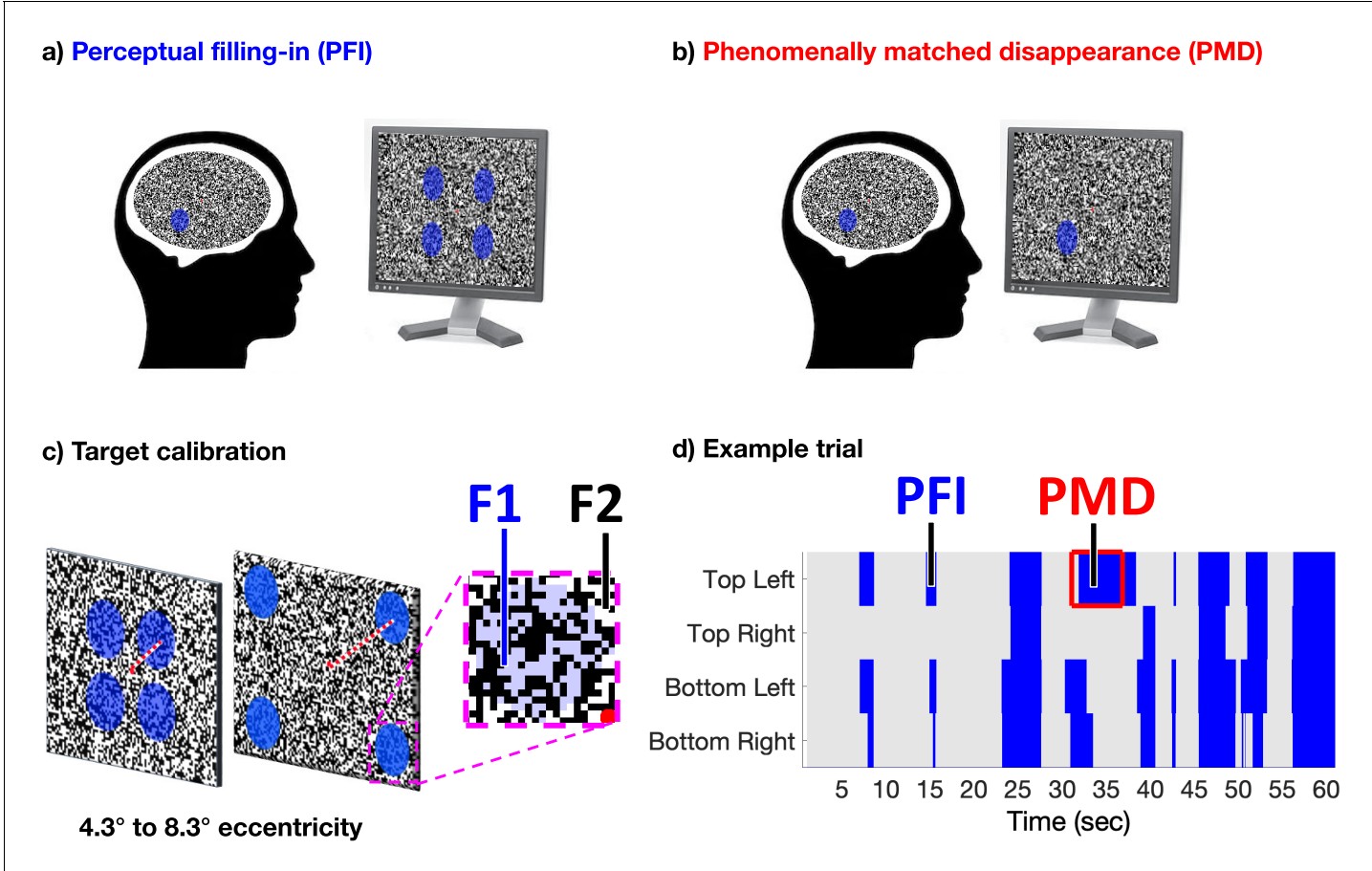

**Figure 1.** Task design. (a) Four peripheral targets, defined by their blue-color, circular shape, and temporal flicker, were superimposed over a dynamic texture to induce perceptual filling-in (PFI). (b) Phenomenally matched disappearances (PMD) were embedded within each trial to mimic the subjective quality of PFI, during which one to four targets were physically removed from the screen. (c) The eccentricity of targets was calibrated per participant, using a custom brain-computer-interface, real-time SSVEP display (see Materials and methods). This procedure aimed to find optimal conditions to induce PFI, and large target (f1) and surround (f2) frequency-tagged responses. (d) Each of 48 trials lasted 60 s, during which participants reported on the visibility of all four peripheral targets continuously, using four unique buttons. Blue shading represents button presses in an example trial, during reported target invisibility.

The online version of this article includes the following source data and figure supplement(s) for figure 1:

**Source data 1.** Example trial.
**Figure supplement 1.** Calibrated target eccentricity and excluded subjects.
**Figure supplement 1—source data 1.** Target eccentricity per participant.
**Figure supplement 2.** Behavioral responses during phenomenally matched disappearance (PMD).
**Figure supplement 2—source data 1.** PMD data per participant.

visibility) significantly increased target-SNR (linear-mixed effects, likelihood ratio test: $\chi^2(1)$=11.60, p=6.58 $\times$ 10$^{-4}$; *Figure 2c* -blue), confirming that target-SNR increased as a measure of target invisibility during PFI. Unlike target-SNR, surround-SNR tracked the contents of phenomenology; it increased during the filling-in of targets, compared to their reappearance, from −0.99 to 1.28 s around report ($p_{cluster}$ <0.001; *Figure 2d*), and it linearly increased with the amount of PFI ($\chi^2(1)$ =19.63, p=9.41 $\times$ 10$^{-6}$; *Figure 2f* -black). Thus, target and surround signals are positively correlated during PFI, and positively correlated with the fluctuating number of invisible targets. This result is inconsistent with prior proposals that SSVEP strength is a proxy for target visibility, but it remains consistent with the proposal that SSVEP strength may be a marker of attentional amplification.

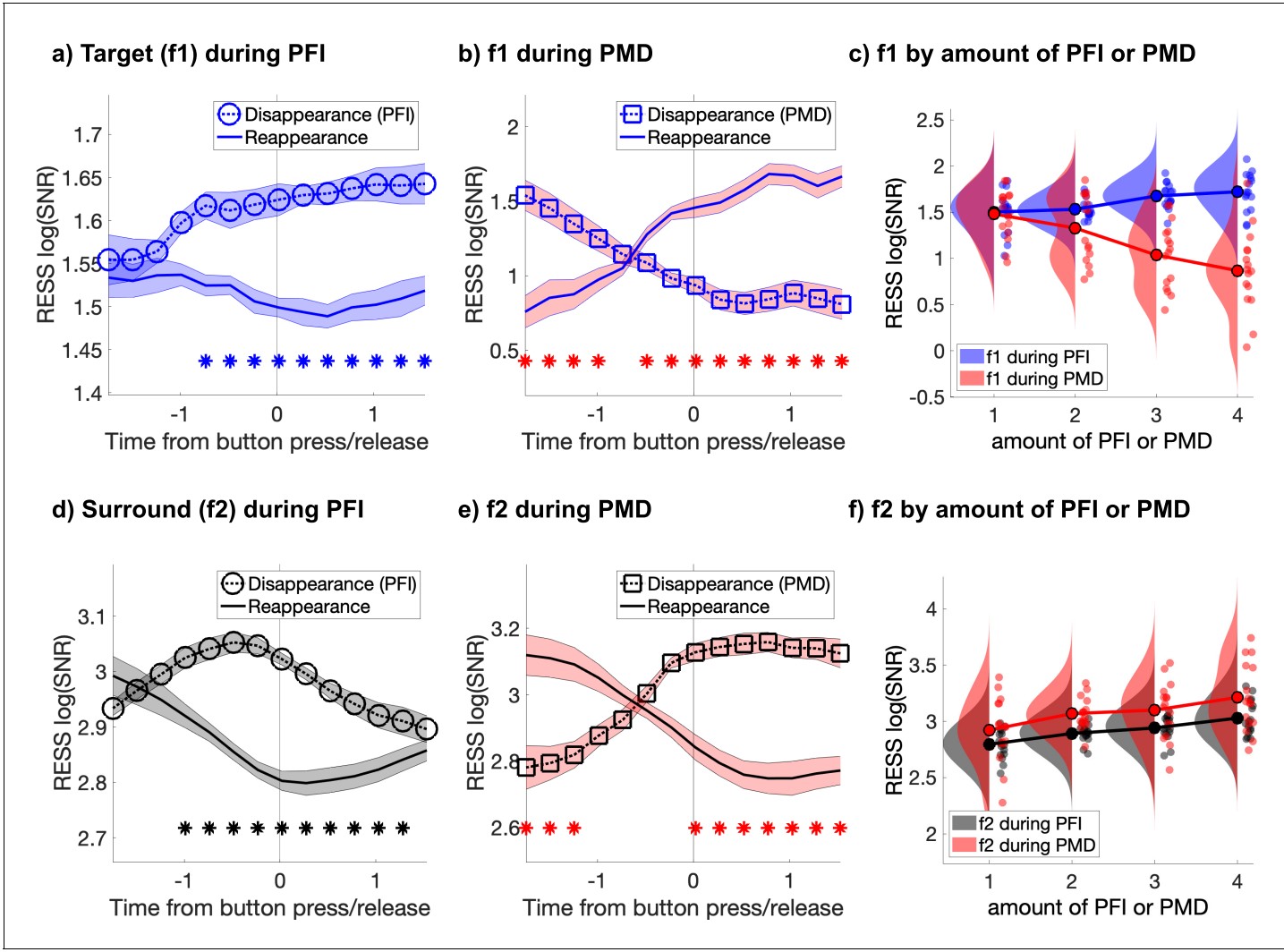

**Figure 2.** Target and surround-specific neural responses during PFI and PMD. (**a**) During PFI target-SNR negatively correlates with target visibility (i.e. increases at disappearance and decreases at reappearance). (**b**) During phenomenally matched disappearance (PMD), when targets were physically removed, target-SNR positively correlates with target visibility. (**c**) Target-SNR during PFI and PMD are in opposition, showing positive and negative correlations with trial-by-trial fluctuations in the amount of invisible targets (defined as the duration-weighted number of visible targets; see Materials and methods). (**d and e**) Surround-SNR increases during PFI (**d**), and PMD (**e**), matching phenomenology as targets become interpolated by their surroundings. (**f**) Trial-by-trial fluctuations in the amount of PFI and PMD positively correlate with surround-SNR. Shading in a, b, d, e corresponds to 1 SEM corrected for within-subject comparisons (*Cousineau, 2005*). Asterisks mark significant differences between the disappearance and reappearance SNR time-courses ($p_{cluster}$ <0.001).

The online version of this article includes the following source data and figure supplement(s) for figure 2:

**Source data 1.** SNR during PFI and PMD.

**Figure supplement 1.** Behavioral data comparing PFI characteristics based on the number of targets perceptually filled-in (nPFI).

**Figure supplement 1—source data 1.** Observed and shuffled data per participant.

**Figure supplement 2.** Whole trial SSVEP results.

**Figure supplement 2—source data 1.** Whole trial power and SNR per participant.

**Figure supplement 3.** Preprocessing for event-by-event based image analyses.

**Figure supplement 3—source data 1.** Example PFI events.

## Increased SNR during invisibility is unique to targets during PFI

We also analyzed the time-course of SNR during phenomenally matched disappearance (PMD) periods (*Figure 1b*). These were events embedded within every trial, during which the phenomenal disappearance experienced during PFI was mimicked by physically removing targets from the screen

(see Materials and methods). During PMD, target-SNR significantly decreased ($\chi^2$(1)=18.35, p=1.84 $\times$ 10$^{-5}$; *Figure 2c* -red), rather than increased as during PFI. No dissociation between PMD and PFI periods was observed for surround-SNR, which significantly increased with increasing amounts of PMD ($\chi^2$(1)=7.37, p=0.007; *Figure 2f* -red), as it did for PFI. This pattern of results confirms that only during PFI, when periods of invisibility were endogenously generated, did the specifically target-sensitive neural responses increase in SNR.

## Visibility negatively correlates with a neural measure of attention

An increased SSVEP response to invisible targets is incompatible with the long-standing interpretation that SSVEP amplitude indexes stimulus visibility, or the contents of consciousness. Increased SSVEP responses have also been reported to reflect changes in arousal (*Keil et al., 2003*), demands during working memory maintenance (e.g. *Ellis et al., 2006*; *Perlstein et al., 2003*; *Silberstein et al., 2001*), and tonic states of vigilance (*Silberstein et al., 1990*) among other factors (reviewed in *Norcia et al., 2015*; *Vialatte et al., 2010*). Of these possible explanations, we considered one of the likeliest candidate interpretations, in this experimental paradigm in particular, to be that SSVEP responses primarily reflect fluctuations in the allocation of attention (*Mora-Cortes et al., 2018*). Indeed, attention increases the likelihood of PFI in psychophysical experiments (*De Weerd et al., 2006*; *Lou, 1999*) and SSVEP amplitude has been extensively validated as a measure of attention to a flickering stimulus (*Ding et al., 2006*; *Hillyard et al., 1997*; *Kashiwase et al., 2012*; *Kim et al., 2007*; *Morgan et al., 1996*; *Müller et al., 2006*, *Müller et al., 1998*; *Müller and Hillyard, 2000*; *Müller and Hübner, 2002*; *Steinhauser and Andersen, 2019*; *Toffanin et al., 2009*; *Walter et al., 2012*).

To test these possibilities, we performed an exploratory analysis, in which we used evoked responses in the alpha band (8–12 Hz), baseline corrected from −2.5 to −1.5 s prior to button press during PFI (*Figure 3*). Event-related reductions in alpha are commonly interpreted as a measure of enhanced cortical excitability under top-down control (*Halgren et al., 2019*; *Jensen and Mazaheri, 2010*), and correlate with the subjective intensity of attentional effort (*Macdonald et al., 2011*). Evoked alpha responses are thus frequently regarded as a measure of attention (*Foxe et al., 1998*; *Foxe and Snyder, 2011*; *Gould et al., 2011*; *Sokoliuk et al., 2019*; *Worden et al., 2000*). Accordingly, we hypothesized that an event-related reduction in alpha, and contribution to the changes in SSVEP strength we have observed, would be consistent with a role of attention during PFI.

With a cluster-based permutation test (*Maris and Oostenveld, 2007*; see Materials and methods), we confirmed a significant difference between baseline alpha amplitude (averaged −2.5 to −1.5 s prior to button press), and alpha amplitude evoked during PFI (*Figure 3b*; $p_{cluster}$ = 0.004, electrode-time cluster corrected). This condition difference corresponded to a centro-parietal cluster of electrodes, and reduction in alpha during PFI from −0.14 s through to 0.65 s after button press. During PMD, a smaller centro-parietal cluster of electrodes, and briefer later latency range (0.35 to 0.41 s) was significant (*Figure 3c*; $p_{cluster}$ = 0.006, electrode-time cluster corrected).

To grasp a clearer picture of the significant effect we identified, we also display the time-course of the evoked reductions in alpha during PFI and PMD, averaged over the electrode clusters depicted in *Figure 3b and c*, respectively. When comparing the evoked alpha during PFI and PMD to the prestimulus baseline, alpha amplitude significantly reduced from approximately −0.77 to 1.58 s during PFI (*Figure 3d*; p<0.011 at FDR q = 0.05). During PMD, no significant reduction in alpha was retained after false-discovery rate corrections (FDR q = 0.05). Across participants, mean evoked alpha suppression also negatively correlated with the strength of target-SNR during PFI (r = −0.58, p=0.019, *Figure 3e*). No correlation was found between evoked alpha reductions and target-SNR during PMD (r = −0.34, p=0.22).

Overall, our analyses demonstrate a sustained decrease in alpha power that was unique to PFI, and that correlated with the strength of target-SNR enhancement. Importantly, the most parsimonious explanation for this convergent evidence is that during PFI, when selective attention is drawn to the target, the strength of target SSVEPs is enhanced. We consider other explanations of the alpha suppression that involve global arousal, vigilance, and motor preparation in our Discussion.

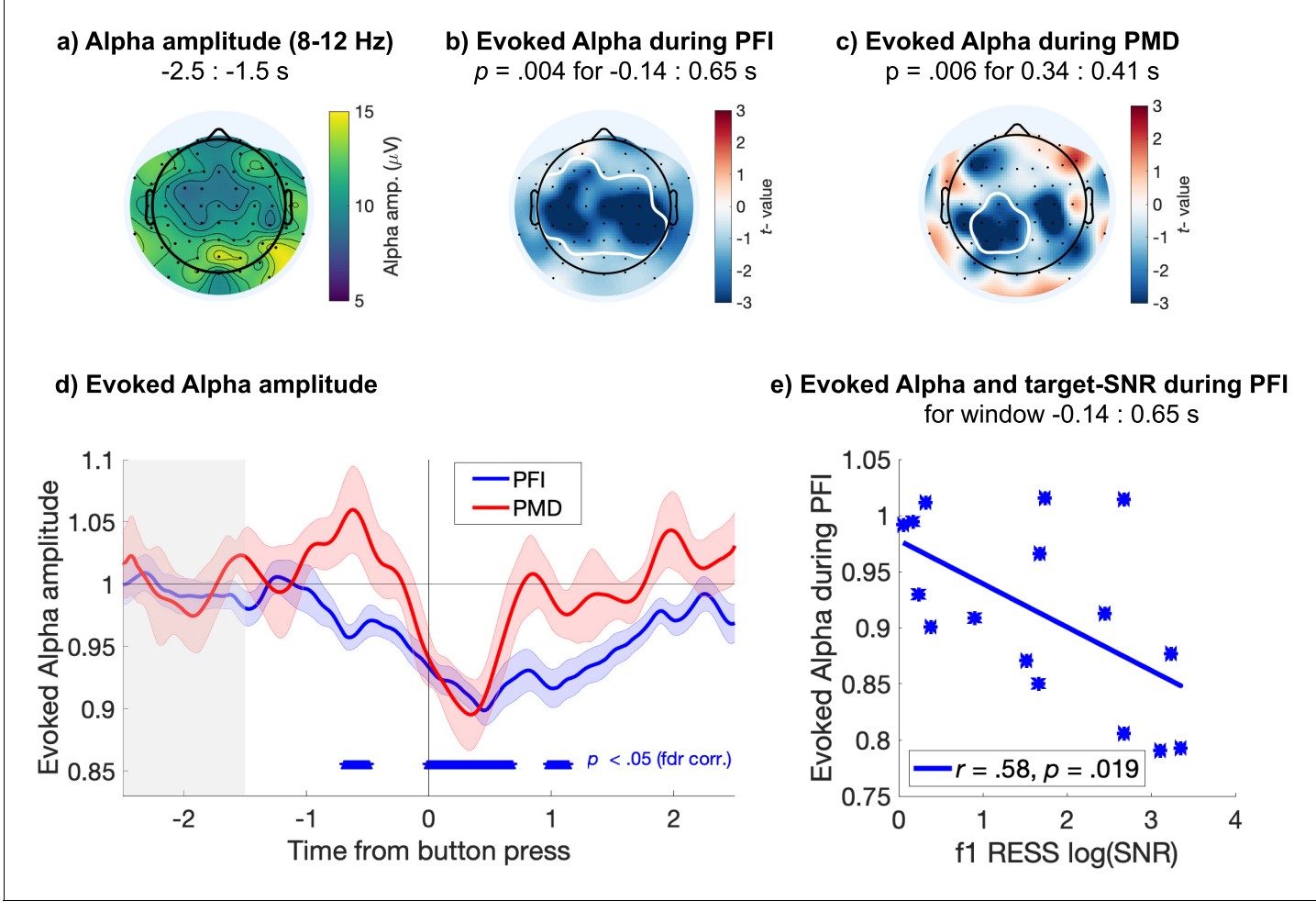

**Figure 3.** Evoked responses in the alpha band. Centro-parietal alpha amplitude decreases during PFI. (**a**) Spatial topography of alpha band (8–12 Hz) amplitude during baseline (−2.5 to −1.5 s before button press). (**b**) PFI- and (**c**) PMD- evoked reductions in alpha amplitude, with significant clusters outlined in white (electrode-time cluster corrected). (**d**) Time course of evoked alpha during PFI (blue) and PMD (red), averaged over the significant electrodes marked in b and c, respectively. Shading shows 1 SEM, corrected for within-participant comparisons (*Cousineau, 2005*). Blue asterisks mark the significant time-windows for the evoked alpha during PFI, averaged over significant electrodes in b (p threshold = 0.011 at FDR q = 0.05). The effect during PMD did not survive FDR corrections for multiple comparisons. (**e**) Each participant's alpha amplitude averaged over the significant time-window shown in b (−0.14 to 0.65 s), plotted against the target SNR at f1 during the same time-window.

The online version of this article includes the following source data for figure 3:

**Source data 1.** Alpha during PFI and PMD.

## Changes in surround-SNR precede target-SNR during PFI

To explain the phenomenon of PFI, the most prominent model proposes that interactions in the early retinotopic cortex are responsible for disappearances (*Pessoa et al., 1998*; *Spillmann and De Weerd, 2003*, for rejections on philosophical grounds, see *Dennett, 1991*; *O'Regan, 1992*; *O'Regan and Noë, 2001*). In this model, a primary stage of seconds-long adaptation occurs in neural populations that are sensitive to the boundary between a target and background region. This stage interacts with an interpolation of the surrounding visual features into the target region, likely achieved via the lateral spread of neural activity through horizontal connections between pyramidal cells (for review; *Spillmann and De Weerd, 2003*). Evidence for this account has primarily been drawn from careful psychophysical manipulations which vary the strength of boundary representations and speed of adaptation within filling-in paradigms (*De Weerd et al., 1998*; *Paradiso and Nakayama, 1991*; *Ramachandran and Gregory, 1991*), as well as single unit recordings in the receptive fields of filled-in regions in monkeys (*De Weerd et al., 1995*). In the latter, De Weerd

et al., first established the average time to experience filling-in when viewing a gray square region on a dynamic texture using four human subjects. Then, using the same apparatus, monkey single-unit responses were shown to increase after several seconds to match the spike rate measured within the same receptive field when viewing only the homogenous texture. As the timing of this climbing activity corresponded to the average timing of filling-in reported in humans, this has been inter-preted in support of an active and early retinotopic filling-in mechanism. The correspondence remains suggestive, however, because these previous studies did not establish a tight correlation between neural activity and perceptual reports through trial-by-trial fluctuations or inter-individual variance.

By uniquely frequency-tagging both the target and surrounds, our paradigm is able to provide new evidence in support of an interaction between the target and background regions, time-locked to behavioral reports in human participants, whose neural activity and perceptual reports were obtained simultaneously. Previous work has analyzed SSVEPs evoked only by a single target during PFI (*Weil et al., 2007*), or only by the surrounding background (*Davidson et al., 2020*), leaving the precise sequence of these neural activations unclear.

Here, we are able to investigate this sequence, and as a consequence of having both target and surround tagging frequencies, look at interactions between the stimulus representations. Due to nonlinearities in visual processing, the interactions between simultaneously presented flickering stim-uli produce nonlinearities in the frequency domain of EEG spectra, known as intermodulation (IM) components (e.g. f2-f1; for review *Gordon et al., 2019*). These IM components have been pro-posed to index interactions between stimulus representations (*Ales et al., 2012*; *Alp et al., 2016*; *Boremanse et al., 2013*; *Gordon et al., 2017*; *Gundlach and Müller, 2013*), which peak prior to a perceptual change during binocular rivalry (*Katyal et al., 2016*), and which increase when stimuli are attended (*Gordon et al., 2017*; *Kim et al., 2017*; *Kim and Verghese, 2012*). Accordingly, we tested whether the relative timing differences in target, surround, or IM-SNR could reveal insights into the temporal relationship between the target and surround representations during PFI.

As *Figure 2* shows, after overlaying the SNR time courses for perceptual disappearance and reap-pearance, the latency of significant changes to target (f1) and surround (f2) SNR differed substan-tially. Specifically, changes to surround-SNR preceded changes to target-SNR by approximately 200 ms (one time-step in our moving window spectrogram analysis). We also observed that the magni-tude of IM-SNR transiently peaked prior to a perceptual disappearance, although the overall strength of the IM component was substantially weaker (*Figure 4—figure supplement 1*). Next, we proceeded by analyzing the relative timing of these differences in SNR at f1, f2, and IM, by retaining the first significant time-point in a temporally contiguous cluster (see Materials and methods). In order to estimate the reliability of these timing differences, we performed a non-parametric jackknife resampling procedure, repeating our analysis in a leave-one-out process. The median, jackknifed estimates replicated the pattern observed in our group-level data (*Figure 4*). During PFI, the median first significant time-point for f2 (median = −1.22 s, SD = −0.02) preceded f1 (median = −0.43 s; SD = 0.26 s). Importantly, for every jackknifed subsample, changes in f2 always preceded f1, and the difference between these distributions was significant (paired *t*-test, *t*(15) = 9.36, p=1.94 × 10$^{-7}$; Wil-coxon signed-rank test, z = 3.52, p=4.38 × 10$^{-4}$). The timing of changes in IM-SNR was not as statis-tically reliable. No significant cluster was retained on 43% of our jackknifed subsamples, which we attribute to the reduced strength of IM-SNR. Notably, however, and despite this reduced strength, changes in IM SNR occurred between the times observed for the fundamental frequencies (median = −0.83 s, SD = 0.09).

During PMD periods, the pattern between the target and surround-SNR was reversed. The median first significant time-point for target-SNR (median = −0.52, SD = 0.02) preceded surround-SNR (median = −0.10, SD = 0.10). This reversal was present on every jackknifed subsample (paired *t*-test, *t*(15) = −16.64, p=4.48 × 10$^{-11}$; Wilcoxon signed-rank test, z = 3.52, p=4.38 × 10$^{-4}$). Unlike during PFI periods, changes to the magnitude of IM-SNR occurred after button press (median = 0.73, SD = 0.20), and after changes to the target and surround-SNR on every subsample.

Visual inspection of *Figure 4* reveals that the SNR change latencies for f1 target-SNR are similar between PFI and PMD. To formally test this, we compared the latency of SNR change-points between PFI and PMD conditions. Indeed, target-SNR latencies were equivalent between PFI and PMD conditions (*t*(15) = −1.56, p=0.14), occurring 500 ms prior to button press. This strongly

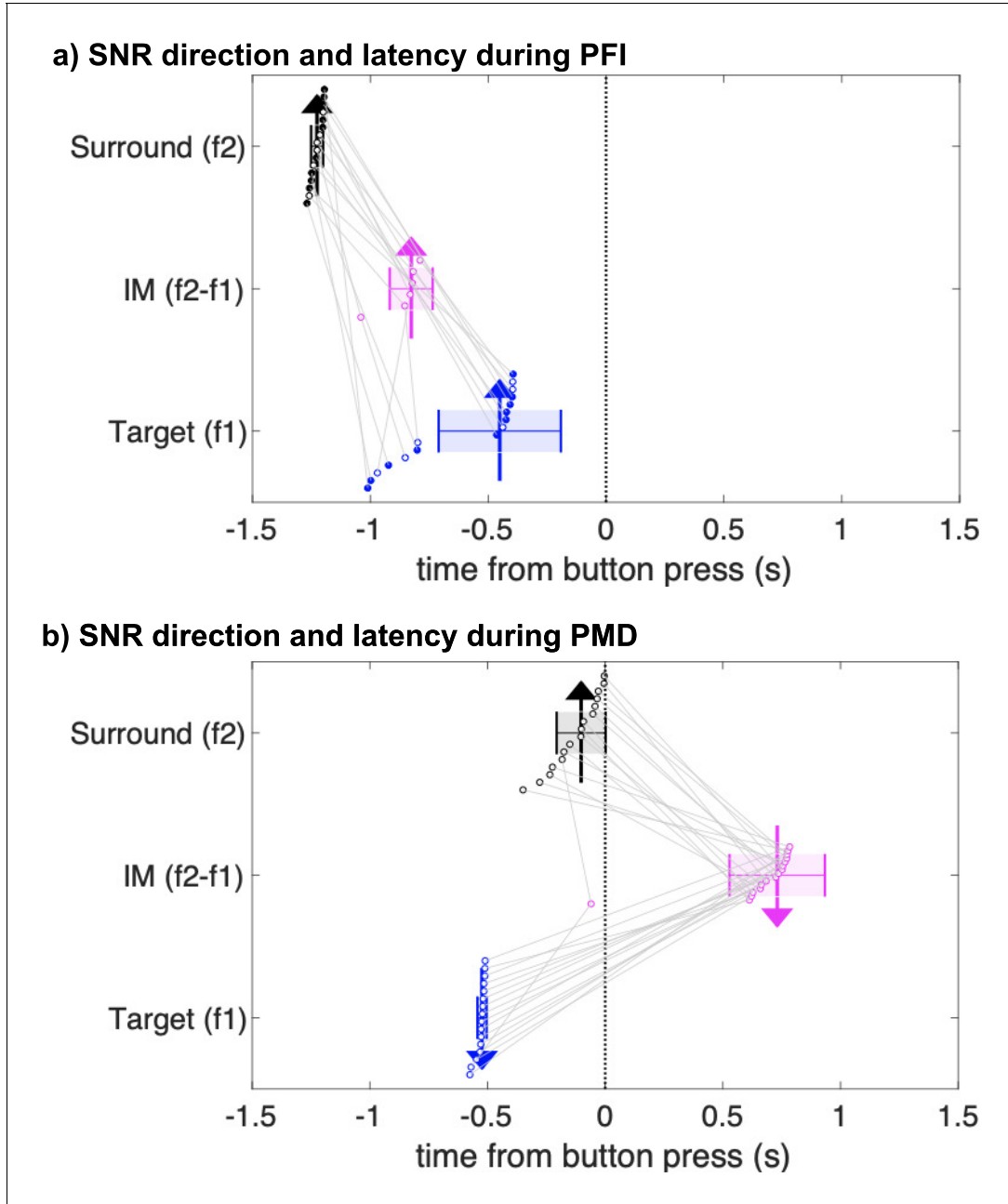

**Figure 4.** Jackknifed latency estimates of SNR timing differences during PFI and PMD. First significant time-points (see Materials and methods) in surround-specific SNR (f2; black), target-specific SNR (f1; blue), and the intermodulation component (IM; pink). Each dot and gray lines link the first significant time-points of each instance of 16 jackknife analyses. The x-position of the arrows indicates the median of these time-points. The direction of the arrows indicates the direction of SNR changes when comparing disappearance to reappearance events. Upward arrows indicate SNR increased during target disappearance; downward arrows indicate SNR was greater during reappearance. Colored shading and whiskers indicate the standard deviation of jackknifed estimates. Note that the IM time-course did not reach significance and is not defined for some subsamples during PFI as indicated by filled black and blue dots for the surround and target.

The online version of this article includes the following source data and figure supplement(s) for figure 4:

**Source data 1.** PFI and PMD change latencies.
**Figure supplement 1.** Time course of SNR of the IM (f2-f1: 5-Hz)during PFI and PMD.
**Figure supplement 1—source data 1.** IM SNR during PFI and PMD.

suggests that target-SNR changes close to the time of perceptual reports, in stark contrast to the surround-SNR, and IM. The distribution of these subsampled estimates is displayed in *Figure 4*.

## Discussion

We designed a PFI paradigm which, combined with our novel dynamic texture tagging method, allowed us to measure sizable SSVEPs attributable to both targets and their surroundings while targets reliably faded from consciousness. We asked participants to report on the disappearance of multiple visual stimuli and revealed that the SSVEP strength of disappearing targets *increases* as they become invisible. Based on alpha-band activity, we propose that SSVEPs are highly unlikely to be a precise measure of the conscious perception of a tagged object, as has previously been assumed. Instead, we propose that SSVEPs in this paradigm are more likely to measure neural processes related to attentional enhancement and suppression. We discuss alternative interpretations below, and these results in the context of existing literature distinguishing visual attention and conscious perception, the insights gained regarding neural mechanisms of PFI, and relevance to theories of consciousness.

### Visual attention and conscious perception

The nature of the relationship between attention and consciousness is one of the most hotly debated and empirically examined topics in consciousness research (*Dehaene et al., 2006*; *Koch and Tsuchiya, 2007*; *Tsuchiya and Koch, 2016*; *van Boxtel and Tsuchiya, 2015*; *van Boxtel et al., 2010a*). As reviewed recently (*Maier and Tsuchiya, 2020*), increasing psychological and neuroscientific evidence suggests that attention and consciousness are not supported by the same mechanisms (see also *Norman et al., 2013*; *Wyart and Tallon-Baudry, 2008*; *Wyart and Tallon-Baudry, 2009*). Among this accumulating evidence, counterintuitive findings that attention can actually decrease visual awareness in certain situations provide the most direct psychological evidence to argue for the dissociation between attention and consciousness (e.g. *Ling and Carrasco, 2006*; *Lou, 1999*; *Mehrpour et al., 2020*; *van Boxtel et al., 2010b*; *Yeshurun and Carrasco, 1998*). Along with other perceptual phenomena, PFI is among the most promising paradigms to examine these opposing effects of attention and consciousness (*Maier and Tsuchiya, 2020*; *Tsuchiya and Koch, 2016*). Building on this prior research, our paper is the first to demonstrate a neural correlate of such counterintuitive and opposing effects of attention and consciousness.

We examined the neural mechanisms underlying PFI, focusing on the SSVEP (*Norcia et al., 2015*; *Vialatte et al., 2010*). The SSVEP technique is a highly versatile tool, allowing researchers to frequency-tag various psychological processes, such as spatial and feature-based attention (*Morgan et al., 1996*; *Müller et al., 2006*, *Müller et al., 1998*; *Müller and Hillyard, 2000*; *Müller and Hübner, 2002*; *Walter et al., 2012*) as well the conscious perception of stimuli (*Brown and Norcia, 1997*; *Lansing, 1964*). As we and others have previously pointed out, however, perceptual paradigms that manipulate conscious perception tend to confound the effects of attention (*Aru et al., 2012*; *Koch and Tsuchiya, 2007*; *Tse et al., 2005*; *van Boxtel and Tsuchiya, 2015*). Thus, previous uses of the SSVEP technique might have reflected the neural correlates of any or a combination of these factors.

More specifically, when used together with binocular rivalry, SSVEP strength has typically been interpreted as a neural correlate of conscious perception (*Bock et al., 2019*; *Brown and Norcia, 1997*; *Cosmelli et al., 2004*; *Jamison et al., 2015*; *Kamphuisen et al., 2008*; *Katyal et al., 2016*; *Lansing, 1964*; *Lawwill and Biersdorf, 1968*; *Roy et al., 2017*; *Srinivasan et al., 1999*; *Sutoyo and Srinivasan, 2009*; *Tononi et al., 1998*; *Zhang et al., 2011*). In one notable study, a reduction in SSVEP strength was interpreted to indicate the cessation of binocular rivalry entirely (*Zhang et al., 2011*). In their study, Zhang and colleagues obtained this result after withdrawing attention from rivalry stimuli and concluded that 'binocular rivalry requires visual attention' (see also *Brascamp and Blake, 2012* for behavioral evidence for this claim).

Another viable, and potentially simpler explanation, is that SSVEP strength instead reflects attention, which is typically positively correlated with conscious perception. By opposing the effects of attention and conscious perception, and demonstrating that the strength of target SSVEPs do not capture visual phenomenology, our current results are in line with this more parsimonious interpretation.

Importantly, our interpretation that SSVEPs reflect attention but not consciousness is supported by indirect, yet convergent evidence. First, psychophysical studies of PFI have established that increased attention to targets, which share colors or forms facilitates their disappearance (*De Weerd et al., 2006*; *Lou, 1999*). We capitalized on this result by designing targets which shared color, form, and temporal flicker, and replicate that these targets disappear together more frequently than by chance, strongly suggesting an interhemispheric facilitatory grouping mechanism. Second, an event-related suppression of alpha-band activity is consistent with spontaneous increases in attention around the time of PFI (*Figure 3d*). Moreover, the same pattern was not observed for PMD periods, which placed equivalent demands as PFI on reported-related motor selection (*van Ede et al., 2019*), and somatosensory spatial attention (*Anderson and Ding, 2011*), suggesting this alpha-band suppression was not related to the mu rhythm. Third, across participants, the magnitude of target-SNR increase correlated with the depth of alpha suppression (*Figure 3e*), but only during PFI periods. Fourth, our IM results (at f2-f1; *Figure 4*), also indicate a transient increase around the time of PFI disappearance, which peaked between the enhancement of the surround (f2) and target (f1) SNR. Previous work has shown that IM-SNR is strongly dependent upon attentional selection of multiple objects (*Gordon et al., 2019*; *Kim et al., 2017*), which is consistent with attention to the target and surrounds in our PFI paradigm. The same pattern was not observed during PMD, where target-SNR decreased prior to button press, and was followed by changes to the surround and IM-SNR, *after* button press. This delayed modulation of the IM signal compared to PFI periods can be understood by observing that IM during PMD is relatively constant during disappearances, but IM-SNR increases after targets are physically returned to the screen (*Figure 4—figure supplement 1*).

Jointly, these four distinct and convergent lines of evidence argue against other alternative explanations. Other interpretations, such as SSVEP changes due to an increase in global arousal, or vigilance (*Silberstein et al., 1990*), generally occur over tens of seconds, and so seem unlikely to account for the transient and temporally ordered SNR changes we report. Further, these and other alternate accounts, such as motor preparation and somatosensory attention, cannot easily explain why both the target and surround-SNR also decrease when targets reappear during PFI (*Figure 2*), as reporting on disappearance and reappearance seem to require similar levels of global arousal and motor preparation. To explain these target-specific, directionally-specific, and temporally ordered effects, we believe that it is most parsimonious to invoke an explanation that involves a known interplay between attention and PFI, and a transient enhancement of SSVEP responses due to attention, although in the case of PFI, the effect of attentional enhancement is to paradoxically increase the strength of an invisible signal.

## Neural mechanisms of PFI

Existing theories propose that a combination of activity in the retinotopic visual cortex (*De Weerd et al., 1998*; *De Weerd et al., 1995*), and higher-level association areas (*Dennett, 1991*; *De Weerd et al., 2006*; *Kingdom and Moulden, 1988*; *Pessoa et al., 1998*) initiate and maintain periods of invisibility during PFI (*Weil and Rees, 2011*). Here, we demonstrate that an increase in surround-SNR precedes an increase in target-SNR by approximately 200 ms (*Figure 2*) and that the temporal order of these activations is consistent across participants (*Figure 4*). Moreover, the timing of significant changes to the intermodulation (IM) component falls between those for the surround- and target-SNR (*Figure 4*). The same pattern was not replicated for PMD periods, as first the target, then surround, and finally IM-SNR were significantly modulated.

Broadly speaking, IM components have been interpreted as an index of the non-linear integration of neural signals, and the existence of neural interactions between sensory stimulus representations (for review see *Gordon et al., 2019*). For example, changes in IM strength have previously been observed when two visual stimuli compete for perceptual dominance (*Katyal et al., 2016*; *Sutoyo and Srinivasan, 2009*; *Zhang et al., 2011*), and to peak prior to a change in conscious percept during binocular rivalry (*Katyal et al., 2016*). As a consequence, we view the existence and temporal order of IM activations to reveal an important piece of evidence with respect to the neural mechanisms of PFI in human participants. This temporal ordering effect seems difficult to reconcile with a model of PFI, which claims that interactions between stimulus representations are unnecessary to account for the phenomenology of filling-in (*Dennett, 1991*; *De Weerd et al., 2006*; *Kingdom and Moulden, 1988*; *Pessoa et al., 1998*).

However, elements of our stimulus design also challenge features of the low-level model. Notably, one key component of low-level accounts is that disappearance is preceded by an adaptation of neurons sensitive to the border between a target and surround. For example, when an achromatic target with a more clearly defined border is filled-in, neural correlates in human V1 and V2 have been identified with fMRI, and BOLD responses reduce in target-sensitive regions in these cases (*Mendola et al., 2006*; *Weil et al., 2008*). In our display, target borders were continuously changed due to the dynamic texture that was shared by targets and surrounds (*Figure 1c*). As a consequence, boundary adaptation may be reduced (*Spillmann and Kurtenbach, 1992*), and it remains possible that neurons higher in the visual hierarchy with a sensitivity to color (such as in V4) are the neural correlates of PFI in our paradigm.

## Theories of consciousness and visual phenomenology

At first sight, our finding of neural activity which increases in magnitude when a target stimulus becomes invisible, appears incompatible with various theories of consciousness. For example, first-order theories of conscious perception (*Mehta and Mashour, 2013*; *Zeki, 2007*) propose that the strength of activity within content-specific visual processing areas determines the vividness of that visual experience. Similarly, the global neuronal workspace theory (GNWT) of consciousness (*Dehaene and Naccache, 2001*; *Mashour et al., 2020*) also considers stronger neural activation as a prerequisite to entry within the global neuronal workspace and subsequent conscious access (*van Vugt et al., 2018*). Insofar as the strength of target SSVEP signals when measured with EEG may reflect the strength of target-specific spike activity in the sensory cortex, our finding does challenge such theories that propose response strength is positively correlated with the vividness of conscious experience. Currently, however, the exact mechanisms of SSVEP generation are not well understood (*Kawashima et al., 2020*; *Kim et al., 2017*; *Norcia et al., 2015*; *Notbohm et al., 2016*; *Rager and Singer, 1998*; *Salelkar and Ray, 2020*), and future studies will be necessary to clarify their input.

It is important to acknowledge the precedent in the literature of a negative correlation between the magnitude of neural spiking and visibility. Prominent examples include single neuron spike-rates recorded during binocular rivalry (reviewed in *Logothetis, 1998*). In these studies, each neuron's receptive field and preferred stimuli was carefully mapped before a binocular rivalry session in macaques. During recording, 22% of the recorded neurons in V4 and MT were responsive during rivalry. Of these neurons, half showed enhanced firing rates when the preferred stimulus was dominant, while the other half showed enhanced firing rates when their preferred stimulus was *suppressed* (*Logothetis, 1989a*; *Logothetis and Schall, 1989b*). These neurons in anti-correlation with the animals' perception counter the notion that the strength of activity early in the sensory cortex determines the contents of conscious perception.

Meanwhile, there exists another family of theories that propose that certain types of causal interactions underlie conscious phenomenology, such as recurrent processing theory (*Lamme, 2006*; *Lamme, 2014*; *Lamme and Roelfsema, 2000*) and integrated information theory (*Haun and Tononi, 2019*; *Tononi, 2004*). For these theories, it is not the strength, but the patterns of neuronal connectivity and their states that determine phenomenology as a whole. To test these theories, consciousness research may need to move away from simple detection or discrimination paradigms, where the visibility of an isolated object is measured against a uniform background. Instead, enriching the sensory input with multiple features and a meaningful background could provide empiricists with a richer repertoire to investigate the interplay of neurons in the visual cortex, and, for example, to investigate how the spatial structure of vision arises (*Haun and Tononi, 2019*; *Koenderink et al., 2014*).

Here, by introducing a dynamic texture tagging display, we have taken a first step toward characterizing the interactions between various visual features based on the strength of SSVEP signals. To tease apart the neural correlates of conscious phenomenology from those of attention and reports, future work may be able to test whether SSVEPs during PFI at attended/unattended locations replicates the pattern which we have described, using innovative no-report methods to time-lock event-related PFI at unattended locations (*Tsuchiya et al., 2015*).

## Summary

We show increased neural responses to the disappearance of stimuli, which challenges long-held beliefs regarding a positive correlation between the strength of SSVEPs and visual phenomenology. From indirect, yet convergent evidence, we conclude that SSVEP strength in our paradigm is more parsimoniously explained to reflect a neural correlate of attention. This demonstrates the need to carefully re-examine previous uses of the SSVEP as a neural correlate of consciousness, and positions the dynamic texture tagging technique as a promising method to deepen our understanding of SSVEP mechanisms, and open new lines of empirical enquiry in consciousness research.

## Materials and methods

### Participants

Nineteen healthy adults (12 females, 19–40 years, $M$ = 26.95, $SD$ = 7.63) with normal or corrected-to-normal vision participated in this study. They were recruited via convenience sampling from students at Monash University, provided written informed consent prior to taking part, and were paid 20 AUD per hr of their time (approximately 3 hr total). Participants with self-reported sensitivity to flickering stimuli were excluded. Ethics approval was obtained from the Monash University Human Research Ethics Committee (MUHREC #CF12/2542 - 2012001375).

### Apparatus and stimuli

Participants sat approximately 50 cm from a computer monitor (size 29 × 51 cm, resolution 1080 × 1920 pixels, subtending 32 × 54° visual angle, refresh rate 60 Hz). The background surrounding our target stimuli (*Figure 1*) was refreshed at a rate of 20 Hz by randomly selecting one of 100, pre-calculated, random patterns generated at the start of the experiment. The random patterns were made up by dividing the screen into squares of 10 × 10 pixels (0.3 × 0.3° visual angle), with each square's luminance randomly set to either black or white. Visually, this procedure appeared similar to random noise, with an equal proportion of black and white texture. At the centre was a 5 mm (0.57° visual angle) diameter red dot serving as a fixation point.

One circular target was presented in each of the four visual quadrants (top left, top right, bottom left, and bottom right). Targets subtended 6.08° visual angle in diameter, within which the white background squares were instead colored as blue/purple (RGB value of 205, 205, 255). There were no line-contours at target boundaries, as target boundaries were defined by the pixelated squares of the background they were overlaid on (*Figure 1c*). As a result, during PFI, target regions became perceptually indistinguishable from the surrounding background texture. The refresh rate of texture within the target regions was set to 15 Hz, with target eccentricity individually calibrated per participant to evoke optimal PFI and SSVEP strength (see *Figure 1c*, and Pre-experiment SSVEP calibration of target eccentricity).

### EEG acquisition

Throughout each session, EEG was recorded with 64 active electrodes arranged according to the international 10–10 system (BrainProducts, ActiCap RRID:SCR_009443). Electrode impedances were kept below 10 kΩ prior to experimentation, and recorded using the default reference (FCz) and ground electrode (AFz) via Brainvision (RRID:SCR_002356) recorder software (sampling rate = 1000 Hz, offline bandpass of 0.5–70 Hz). All EEG data was stored for offline analysis using custom Matlab scripts (RRID:SCR_001622; Ver: R2016b), as well as the EEGLab (RRID:SCR_007292) (*Delorme and Makeig, 2004*), Chronux (RRID:SCR_005547) (*Bokil et al., 2010*), and Fieldtrip (RRID:SCR_004849) (*Oostenveld et al., 2011*) toolboxes. Prior to experimentation, the BCILAB toolbox (RRID:SCR_007013) (*Kothe and Makeig, 2013*) was also implemented to calibrate real-time SSVEP strength (see below).

### Pre-experiment SSVEP calibration of target eccentricity

Finding optimal conditions for SSVEP and PFI paradigms is fraught with difficulty, as the features which enhance each are in direct opposition: in order to optimize SSVEPs, large and foveally-positioned targets are preferred (*Norcia et al., 2015*), while to optimize PFI, small and peripherally-

positioned targets are required (*Anstis, 1996*; *De Weerd et al., 1998*). We solved this problem using a custom brain-computer-interface (BCI) procedure.

The purpose of the calibration procedure was to find a target eccentricity at which PFI readily occurred, while evoking observable peaks in the EEG spectra at the target flicker frequency (15 Hz). Targets were initially centred close to fixation (4.3° visual angle in eccentricity measured from the fixation dot to the center of the target). Participants were instructed to fixate the central dot and were asked to report on PFI at the parafovea. If participants reported PFI, they were then asked to describe their experience more explicitly. Depending on their subjective reports, we adjusted the eccentricity of the targets along the diagonal (from 4.3 to 8.7°, in steps of 0.4°) so that their perceptual experience was described as 'appearing and disappearing often, for a few seconds at a time'. All participants reported that PFI occurred.

Concurrently with these manipulations, the power spectrum as well as log(SNR) spectrum at POz were displayed in real time. Due to the computational demands of presenting frequency-domain EEG spectra in real time, no inferential statistics were used to define adequate SSVEP strength. Readily observable 15 and 20 Hz peaks in the EEG spectra were taken as evidence of frequency-tagging at face value. In the absence of readily observable peaks in the real time EEG spectra, target eccentricity was reduced, and the process was repeated. If tagging was unsuccessful at all settings ($n$ = 2), the largest and most central target position at which any PFI occurred was adopted. This was under the assumption that repetition over many trials may still result in a frequency-tag, which was confirmed in our analyses. *Figure 1—figure supplement 1* displays the final calibrated target eccentricity across participants ($N$ = 19).

## Experimental procedure

Participants were instructed with the following script: 'Fixate on the red dot. If you perceive that any of the four targets has completely disappeared, press the button corresponding to that target and hold it down for as long as you perceive that target to be absent. If more than one target vanishes simultaneously, try to report on them all as accurately as possible.' Specifically, they were instructed to press keys 'A', 'Z', 'K', and 'M' on a traditional QWERTY keyboard, assigning them to the upper left, bottom left, upper right, and bottom right targets, respectively. After calibration, each participant completed one 60 s practice trial, followed by 48 self-paced, experimental trials which were also one minute long. Participants took mandatory breaks of 3–5 min every 12 trials, while EEG recordings were paused and channel impedances were monitored.

## PMD periods

Each trial included one randomly generated PMD period of between 3.5 and 5 s (drawn from a uniform distribution), during which one to four targets were physically removed from the screen. Participants were not informed of these physical disappearance periods before the experiment. To mimic the frequency of genuine PFI, PMD did not occur in the first 10 s of each trial (*Schieting and Spillmann, 1987*). The order of all PMD periods were randomized for each participant, as were the location of removed targets in the case of one, two, and three targets. These physical disappearance periods enabled us to monitor how well participants were able to report on the visibility of four simultaneously presented targets, and additionally served as a control condition for comparison with the neural signals evoked by genuine PFI.

## Participant and trial exclusion based on PMD periods

We have previously demonstrated that the use of PMD periods can identify participants who are unable to report on four simultaneous targets, as well as experimental trials which are unsuited for further analyses (*Davidson et al., 2020*). In particular, whether a participant accurately reported the PMD period on time (via button press) was used to estimate that participant's engagement with the task.

Unlike our previous study (*Davidson et al., 2020*), we tailored our stimulus so that PFI happens very often across all participants. Thus, it was often the case that the physical removal of a target (PMD onset) occurred for targets which have already disappeared from consciousness due to genuine PFI. Such occurrences were more frequent for those participants experiencing greater amounts of PFI. To obtain accurate estimates of how accurately and quickly participants responded to the

physical removal of a target we estimated the baseline button-press likelihood per individual participant, by performing a bootstrapping analysis (with replacement). To perform this bootstrap, for any PMD onset in trial T at time S (seconds), we randomly selected a trial T' (T = T' was allowed) and we epoched [S-2, S+4] button press data at corresponding PMD target locations in T'.

We repeated this for all available trials (T = 1...48) to obtain a single bootstrapped set of 48 trials per participant. This procedure was repeated 1000 times, and the mean button press time-course for each 48-trial set was retained as the null-distribution of button-press likelihood, over time, per participant. *Figure 1—figure supplement 2* (gray) shows the bootstrapped likelihood of button press for a single participant. We then used the null distribution z-scores of ±1.96 as the 95% CI for each participant (after first converting the data using the logit transformation due to a violation of normality). We defined the reaction time to PMD-onset as the first time point after which each participant's median button-press data exceeded the top CI of their null-distribution likelihood. We excluded participants (*n* = 3, See *Figure 1—figure supplement 1*) with no PMD-onset reaction time in the first 2 s (i.e. [0, S+2]). We note that for two of these participants, it appeared that buttons were consistently released at PMD-onset – potentially indicating buttons were released during PFI rather than pressed as per instructions. For the remaining participants, the mean reaction time to respond to PMD onsets, and thus the disappearance of a peripheral target was 0.68 s (*SD* = 0.31).

Having identified which participants could successfully indicate target disappearance based on their button press data (*N* = 16), we continued to remove any trials in which a PMD was not correctly detected from subsequent analysis. This ensured that throughout each trial, participants were accurately reporting on PFI, and that all retained data was on task. To identify individual trials for exclusion, we regarded a PMD period as failed if the corresponding button was not pressed for at least 50% of the allowed response time window. This window was the duration of the PMD (3.5 to 5 s). *Figure 1—figure supplement 2c* shows a histogram of the number of rejected trials per participant (overall, only 16 trials were rejected) across 16 retained participants.

## Simultaneous PFI and location-shuffling analysis

To examine whether an increasing number of filled in targets (nPFI; *n* = 0, 1, 2, 3, 4) affected PFI characteristics, we performed a shuffling analysis to create a null distribution that removed the temporal correlation between targets (*Davidson et al., 2020*). This analysis tests whether simultaneous PFI occurs more often that can be expected by chance. Specifically, we created 1000 shuffled trials per participant, that could contain the button-press data for each location from any trial throughout their experimental session (e.g. one shuffled trial could consist of top left button press time course from trial 10, top right from trial 43, bottom left from trial 12, bottom right from trial 2). We selected a random trial for each location, allowing multiple locations from one trial to be included to form one shuffled trial (i.e. bootstrap with repetition). This shuffling procedure conserves the total amount of PFI recorded, while ensuring that the button-press data at a given location could come from any independent trial. If target disappearances during PFI were independent, then destroying the temporal correlation in shuffled data should not matter, and shuffled and experimental data would look similar. We repeated our behavioral analysis (as detailed above) on this shuffled data, with the results displayed in *Figure 2—figure supplement 1*.

We also performed quadratic model-fit analyses to compare the magnitude and direction of the fitted coefficients for observed and shuffled data. For this analysis, we retained the coefficient (β; 2nd order polynomial) from the fit to our observed data as our observed statistic. We also fit the same quadratic model to each of the shuffled data sets (*n* = 1000) and used these coefficient values as a null distribution to compare with the observed β. If the observed β exceeded the top 95% of the null distribution, we regarded the quadratic fit for the observed data as significant at p<0.05.

## EEG preprocessing

After data collection, noisy channels were identified using a modified version of the PREP pipeline (*Bigdely-Shamlo et al., 2015*). We omitted the bad-by-RANSAC criterion that identifies correlated channel groups which deviated from other channels. This was necessary as frequency-tagging elicits responses in localized, often highly correlated channel clusters. Bad channels were then spherically interpolated (channels rejected per participant *M* = 5.70, *SD* = 0.62). After channel rejection and interpolation, whole-trial EEG data were re-referenced to the average of all electrodes, and linearly

detrended before being downsampled to 250 Hz. We then applied a Laplacian transform to improve the spatial selectivity of our data. *Figure 2—figure supplement 2* shows the whole trial spectrum from electrode POz, as well as topographical distribution of frequency-tagged components.

## SSVEP analysis via rhythmic entrainment source separation (RESS)

SSVEP topography can vary based on individual participant anatomy, the entrained neural network, which depends on tasks and stimuli (*Ding et al., 2006*), as well as based on the frequency of flicker selected (as shown in *Figure 2—figure supplement 2*). As such, we applied a spatiotemporal filter called rhythmic entrainment source separation (RESS), to reduce the distributed topographical response at SSVEP frequencies to a single component time-series (*Cohen and Gulbinaite, 2017*). The RESS single component is a weighted average of all channels, which can be analyzed in the time-frequency domain instead of relying upon the selection of a single or multiple channels based on post-hoc data inspection.

Specifically, RESS works by creating linear spatial filters tailored to maximally differentiate the covariance between a signal flicker frequency and neighborhood frequencies, thereby increasing the SNR at the flicker frequency. We constructed RESS spatial filters from 64-channel EEG, by applying a narrow-band Gaussian (full-width at half maximum = 0.5 Hz) filter to the original EEG data in the frequency-domain, centered at the target, surround and IM frequency, respectively (and independently). As the frequency-neighborhood across different signals would contain different amounts of simultaneous flicker, we proceeded by selecting broadband neural activity to construct reference covariance matrices. Comparing signal to broadband activity has previously been shown to allow the reconstruction of SSVEP signals using RESS (*Cohen and Gulbinaite, 2017*; *Davidson et al., 2020*).

From *Figure 2—figure supplement 2*, RESS spatial filters were constructed per participant per frequency of interest (target, surround, and IM). We used epoched data from all time-windows −3000 to −100 ms and 100 ms to 3000 ms around button press/release, avoiding PMD periods. We avoided the 200 ms around button press/release to optimize RESS filters in the absence of motor-evoked activity. Each filter was fitted without distinguishing whether targets were disappearing or reappearing due to button press or release, in order to reduce the possibility of overfitting these condition comparisons. After application of the RESS spatial filters, we reconstructed the time course of SSVEP log(SNR) from the RESS component time-series as described below. With RESS, we were able to focus our analysis on a single component time-series per frequency of interest, without arbitrarily selecting a single channel or averaging channels, eliminating the need for corrections for multiple comparisons across channels.

## SSVEP SNR calculation

In the SSVEP paradigm, we compute the SNR at each frequency, which in logarithmic scale corresponds to log of the power at each frequency minus the mean log power across the neighborhood frequencies. Throughout this paper, the neighborhood used in SNR calculation always excludes the frequency half-bandwidth nearby f Hz, which is calculated with a formula: f = (k+1)/2T, where k = the number of tapers used in time-frequency decomposition and T = temporal window (seconds) of the data.

For our moving-window SNR analyses, we first computed the single-taper (k = 1) spectrogram using a 2.5 s window, with a half-bandwidth of 0.4 Hz, and shifted the window in a step size of 250 ms. We then compared the signal at f Hz to neighborhood [f-2.5, f-0.5] Hz and [f+0.5, f+2.5] Hz. For our whole-trial SNR analysis with a half-bandwidth of 0.017 Hz, we compared the signal at f to neighborhood [f-0.12, f-0.06] Hz and [f+0.06, f+0.12] Hz.

## Evoked responses in the alpha band

We used a two-step analysis process to calculate the relative alpha amplitude during all PFI and PMD periods. First, after all PFI and PMD periods were epoched and preprocessed as described above, we bandpass filtered the time series with a narrow-band Gaussian filter (8–12 Hz, full-width at half maximum = 2 Hz). Second, to the bandpass filtered signals, we applied the Hilbert transform to compute the alpha-band amplitude as the envelope of the Hilbert-transformed filtered signal. Third, to obtain evoked responses in the alpha band, within participants, we divided the time course

of the amplitude by the average over the period −2.5 to −1.5 s prior to button press (gray window shown in *Figure 3d*) for each participant.

To identify electrode-cluster-time points, in which the difference between baseline and evoked alpha responses were significant, we implemented non-parametric cluster-based permutation tests (Fieldtrip *Maris and Oostenveld, 2007*; *Oostenveld et al., 2011*; ft_timelockstatistics.m function). Specifically, using all electrodes, we set a minimum spatial cluster size of three neighbors, and a threshold for cluster identification at p=0.05 (uncorrected). Spatiotemporal clusters of test-statistics (*t*-scores from dependent samples *t*-tests) which exceeded p<0.05 (uncorrected) were then identified. We then corrected for multiple-comparisons using Monte Carlo permutation tests (1000 repetitions). In each permutation, we exchanged the data label between conditions at random within participants, and repeated the above procedure to identify spatiotemporal clusters on these permuted data, and obtained a null distribution for clustered-test statistics (*t*-scores). By comparing the null distribution to the observed cluster statistic, we finally obtain the cluster-level p-value corrected for multiple-comparisons (*Maris and Oostenveld, 2007*).

Cluster-based permutation tests should be interpreted with caution, and used to infer only about the overall statistical contrast, and not to infer the location or latency of a significant effect (*Maris and Oostenveld, 2007*; *Sassenhagen and Draschkow, 2019*). After identifying the presence of a significant cluster, we also displayed the average activity within each cluster, over the entire epoch (−2.5 to 2.5 s after button press). We then compared this activity to baseline and corrected for multiple comparisons using FDR q = 0.05.

## Event-by-event image analysis of button press and log(SNR)

We performed image-based event-by-event analysis (*Fujiwara et al., 2017*), to investigate whether the changes in log(SNR) may reflect the amount of PFI. This image-based analysis is necessary due to variations in the frequency and duration of reported PFI per participant.

To accommodate these differences, we first sorted all PFI events in descending order, based on the sum of buttons pressed over a 3 s period. We used the period [0, +3] relative to button press for PFI disappearances, and [−3, 0] relative to button release for PFI reappearances. This duration-weighted number of buttons pressed we term 'the amount of PFI' (*Davidson et al., 2020*). After sorting based on the amount of PFI, we then resampled participant data along the trial dimension (y-axis) to normalize trial counts to 100 trials for each participant. This process of resampling along the trial dimension was repeated for the event-by-event time course of log(SNR), except the order of trials was predetermined by the corresponding button-press per participant. A schematic pipeline for this entire procedure is displayed in *Figure 2—figure supplement 3*. Finally, to quantify whether changes in log(SNR) occur with an increasing amount of PFI, we grouped trials when the amount of PFI was between 0 and 1, 1 and 2, 2 and 3, or greater than 3. The results have been visualized using the raincloud plots statistical package (*Allen et al., 2019*).

## SNR timing differences

To investigate the timing of changes to target and surround SNR, we superimposed the time-course for disappearances and reappearances, by aligning at button press and release. We then compared the magnitude of these SNR time-courses using consecutive paired samples *t*-tests (two-tailed), at each time point. To confirm the differences between the target and surround-specific changes in SNR, we also performed a non-parametric jackknife resampling procedure. As the waveforms under consideration for f1 and f2 have different peak amplitudes and shapes, we avoided latency estimates based on peak-criterion (cf. *Miller et al., 1998*; *Miller et al., 2009*). Instead, we repeated our temporal cluster-based analysis after subsampling participants in a leave-one-out analysis, to estimate the reliability of this effect. Specifically, we compared the SNR time courses for disappearance and reappearance using running paired samples two-tailed *t*-tests at each time point. The first time point in a temporally contiguous cluster of at least 500 ms was retained as the first difference in SNR during PFI. To increase the temporal resolution of our jackknife estimates, we linearly interpolated the SNR time courses from 250 ms time steps (4 Hz) to 1 ms time steps (1000 Hz).

## Statistical analysis – EEG

To assess the significance of SSVEP peaks in the EEG spectra, we corrected for multiple comparisons with a False Discovery Rate (FDR) q = 0.05 (*Benjamini et al., 2006*; *Figure 2—figure supplement 2*).

We corrected for multiple comparisons in SNR time-series (*Figure 2a,b,d, and e*) using non-parametric temporal cluster-based corrections (*Maris and Oostenveld, 2007*). Specifically, we first detected any temporally contiguous cluster by defining a significant time point as p<0.05 uncorrected. Then, we concatenated the contiguous temporal time points with p<0.05 and obtained a summed cluster-level test statistic for the cluster. The sum of observed test statistics (e.g. *t*-scores) in a temporally contiguous cluster were then retained for comparison with a permutation-based null distribution. To create the null distribution, we repeated the procedure of searching for and retaining contiguous time points which satisfied the p<0.05 (uncorrected) cluster criterion, after first shuffling the condition labels 2000 times. For within-participant comparisons, this amounts to randomly permuting the averages for each condition within each participant. From each of the 2000 repetitions, we obtained the maximum sum of cluster-level test statistics, which served as a null distribution. We regarded the effect to be significant if the original summed cluster-level statistics exceeded the bottom 95% of the null distribution of the summed statistics (as $p_{cluster}$ <0.05).

## Additional information

### Funding

| Funder | Grant reference number | Author |
| --- | --- | --- |
| Australian Research Council | FT120100619 | Naotsugu Tsuchiya |
| Australian Research Council | DP130100194 | Naotsugu Tsuchiya |

The funders had no role in study design, data collection and interpretation, or the decision to submit the work for publication.

### Author contributions

Matthew J Davidson, Conceptualization, Software, Formal analysis, Supervision, Validation, Investigation, Visualization, Methodology, Writing - original draft, Project administration, Writing - review and editing; Will Mithen, Data curation, Investigation, Methodology, Writing - review and editing; Hinze Hogendoorn, Writing - review and editing; Jeroen JA van Boxtel, Naotsugu Tsuchiya, Conceptualization, Resources, Supervision, Funding acquisition, Methodology, Writing - original draft, Project administration, Writing - review and editing

### Author ORCIDs

Matthew J Davidson (iD) https://orcid.org/0000-0002-2088-040X
Jeroen JA van Boxtel (iD) https://orcid.org/0000-0003-2643-0474
Naotsugu Tsuchiya (iD) http://orcid.org/0000-0003-4216-8701

### Ethics

Human subjects: Ethics approval was obtained from the Monash University Human Research Ethics Committee (MUHREC #CF12/2542 - 2012001375). Students at Monash University, provided written informed consent prior to taking part.

### Decision letter and Author response

Decision letter https://doi.org/10.7554/eLife.60031.sa1
Author response https://doi.org/10.7554/eLife.60031.sa2

## Additional files

### Supplementary files

• Transparent reporting form

### Data availability

All data and analysis code has been made available in a repository on the open science framework.

The following dataset was generated:

| Author(s) | Year | Dataset title | Dataset URL | Database and Identifier |
|---|---|---|---|---|
| Davidson M | 2019 | Multitarget PFI - BCI | https://osf.io/hs7fn/ | Open Science Framework, OSF.IO/HS7FN |

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

# Appendix 1

## Supplementary results
### A spatial interaction between target locations increases PFI duration when targets disappear together

We investigated behavioral responses during PFI, and the effect of the number of targets simultaneously invisible. We compared the amount of PFI per trial, duration per PFI, and total duration of PFI during either 0, 1, 2, 3, or 4 target PFI periods (*Figure 2—figure supplement 1*, blue bars). We found that while 0 target PFI periods were most frequent (occurring 6 times per trial), when targets did disappear, simultaneous 4 target disappearances were the most common (4 times per trial, compared to <2 times per trial for 1, 2, and 3 target disappearances). This interesting trend continued for both the duration per PFI (7 s per 0 target, <2 s for 1–3 targets, 5.5 s for 4 targets) and total duration of PFI (32 s for 0, <2 s for 1–3, and 20 s for 4, respectively), showing that 4 target disappearances were the most common, disappeared for a longer duration, and greatest total duration per trial.

*Figure 2—figure supplement 1* (c-h, gray bars) displays the results of this analysis, showing the mean across all 1000 shuffled sets of data. In contrast to observed data, the shuffled data showed PFI for 1, 2, and 3 targets being more common (6, 7, and 6 times per trial respectively), than for 0 and 4 targets (each occurring less than 4 times per trial). For PFI duration, especially durations for 4 target PFI were shorter in the shuffled data. Strikingly, and in direct opposition to our observed data, the total duration of 0, 1, 2, and 3 target PFI per trial was roughly equivalent in shuffled data (each ~14 s duration), with 4 target PFI occurring for the least amount of time (<10 s).

To investigate whether these trends obtained over 4-target locations were likely to occur by chance, we performed a shuffling analysis (see Materials and methods) and recalculated PFI characteristics (PFI per trial, PFI duration, and total duration), as a function of the number of targets filled-in (nPFI).

To statistically evaluate the differences in these trends, we compared the coefficient for the quadratic fit of our observed data to all the coefficients in our null distribution for shuffled data. For all PFI measures, the observed coefficient was outside the 95% of the null distribution (corresponding to $p < 0.05$). For the PFI per trial and total duration per trial, the positive quadratic term for the effect of nPFI in our observed data is in direct opposition to the distribution of quadratic coefficients in our shuffled data. Taken together, we interpret this result as evidence of a synergistic spatial-interaction between multiple PFI targets, that increases the likelihood of simultaneous disappearances across visual hemifields and target locations (*Davidson et al., 2020*).

