## [Decision Letter]

**Acceptance summary:**

Your careful investigation of the neural correlates of visual disappearance during perceptual filling-in (PFI) using steady-state visual evoked potentials (SSVEP) has yielded surprising and insightful results. The finding that SSVEP tracks attention but not visibility during PFI bears important consequences for current theories of consciousness. Congratulations for this interesting article which should stimulate future theoretical and experimental developments.

**Decision letter after peer review:**

Thank you for submitting your article "A neural representation of invisibility" for consideration by *eLife*. Your article has been reviewed by two peer reviewers, and the evaluation has been overseen by a Reviewing Editor and Floris de Lange as the Senior Editor. The following individual involved in review of your submission has agreed to reveal their identity: Benoit R Cottereau (Reviewer #1).

The reviewers have discussed the reviews with one another and the Reviewing Editor has drafted this decision to help you prepare a revised submission.

This manuscript describes a human EEG study which aims at characterizing the neural correlates of visual disappearance during perceptual filling-in (PFI) using steady-state visual evoked potentials (SSVEP). The authors report that target disappearance leads in this paradigm to an increase rather than to a decreased SNR of the target SSVEP. The authors interpret this "neural correlate of invisibility" as an empirical challenge for existing theories regarding the relationship between SSVEP and conscious perception. The two reviewers have found the study to be creative and its findings to be of potential importance for the field. However, they have also raised concerns regarding the interpretation of the findings proposed by the authors, which would require additional analyses to be supported by the data and a more extensive account of the existing literature on the relationship between the neural correlates of visual awareness and attention. There are also concerns regarding the number of subjects included in the analyses which should be clarified. The paragraphs below describe the main concerns that have been discussed among reviewers and the reviewing editor, and that should be addressed explicitly in a revised version of the manuscript.

Essential revisions:

1) Novelty of the findings. Although the findings are interesting and novel in this particular context of perceptual filling-in, they may not be as surprising as the authors present them to be. Indeed, previous work (including some cited by the authors, e.g., Morgan, Hansen and Hillyard, 1996) has shown a clear connection between SSVEP and attention. And it is also known from previous work that PFI increases with attention (Lou, 1999; De Weerd, Smith and Greenberg, 2006). The present finding adds to this existing body of work by showing that when PFI (a "visual disappearance" effect) is used to dissociate awareness and attention, SSVEP follows attention rather than consciousness. In other words, rather than showing that SSVEP is a general neural measure of invisibility, an alternative interpretation of the finding is that SSVEP is a neural measure of attention, which in this particular setting of frequency-tagged PFI becomes a neural measure of invisibility (see also the second concern below).

2) The Abstract and the Discussion should be revised in a way that considers the finding beyond the specific context of PFI used in this study. For example, the suggestion that the "brain actively represents disappearing, or absent stimuli" should be accompanied by the alternative interpretation that SSVEP reflects a neural measure of attention rather than awareness, which in the particular setting of PFI go in opposite directions. Also, as raised by reviewer #1, previous work has studied the dissociation between the neural correlates of visual attention and awareness (among others, see e.g. Wyart and Tallon-Baudry, 2008; 2009; Koivisto et al., 2009; Norman et al., 2013). A deeper review of the existing literature on this topic (in the Introduction and/or Discussion) would afford to better understand what is already known and also to provide insights for future work.

3) Nature of the relation between SSVEP/alpha power and attention. The authors ground their interpretation on previous studies (Lou, 1999; De Weerd et al., 2006) where attending to target features increased PFI probability, and also on the correlation they found between target-tagged SNR and alpha power. However, these are both indirect evidence – in the case of alpha power, it amounts to reverse inference to assign changes in alpha power to changes in attention. And attention was not directly manipulated in the present study, which would have provided a much stronger monitoring of the relation between attention and awareness (see, e.g., Morgan et al., 1996 or Müller et al., 2006). We thus suggest the authors to be more cautious regarding the interpretations that they seek to draw regarding attention in their study. Attention was not manipulated experimentally, and using alpha power as a measure of attention amounts to reverse inference.

4) Specificity of the finding. These results presented in the manuscript do not demonstrate that all sensory-cortical activity goes along with attention instead of awareness, as the authors' Abstract/significance statement/Discussion suggest to be the case ("by showing that the strength of stimulus-specific neural activity can encode the conscious absence of a stimulus", or "we found that neural activity increased […] when targets became invisible"). In Abstract and significance statement, the authors only state "neural activity" or "neural response", instead of SSVEP, which can be misleading. Similarly, in the Discussion, the authors should note that it remains entirely possible that other types of neural activity (e.g. spike rate or recurrent activity) in visual cortex correlates with the vividness of conscious experience, which would in principle be consistent with existing theories of visual awareness (e.g., GNW).

5) Comparison between PFI and phenomenally matched disappearances (PMD). To better characterize the difference between the responses to PFI vs. to PMD, and support the claim that target-tagged SNR decreases rather than increases during PMD (subsection “Increased SNR during invisibility is unique to targets during PFI”), it would be very important to show the target-tagged SNR changes around button press for PMD – i.e. the equivalent of Figure 2B and E for PMD.

6) Neural mechanisms underlying the observed findings. Almost no discussion about the neural mechanisms underlying the surprising finding is provided. It seems important to better explain the cortical processes involved (e.g., the authors could compare more carefully their results with those obtained in macaque electrophysiology by De Weerd et al., 1995). At the analytical level, one possibility would also be to analyse the SSVEP SNR at the intermodulation frequencies (I see in Figure 2—figure supplement 2 that responses at F2 minus F1 = 5Hz are significantly above noise). This would permit to characterize and discuss the interactions between the neural responses corresponding to the processing of the targets and to those corresponding to the processing of the surround (see e.g. Appelbaum et al., 2008).

7) Analysis of SNR change latencies. The analysis of SNR change latencies (currently described in the supplements) would deserve to be more documented and to appear in the main document. The finding that changes in background SNR precede changes in target SNR is an important result which clarifies the temporal sequence of neural activations. It would be interesting to determine when the SNR change corresponding to the inter-modulation product (e.g., at F2 minus F1) appears (see the previous concern).

8) Relation between duration of PFI and SSVEP. In the Discussion, the authors mention: "As more targets disappeared and presumably drew attention, both the duration of their absence and strength of target SNR increased." However, the duration effect, shown in supplementary materials, is not referenced in the main text. In Figure 2, in addition to investigating the relation of SSVEP with the number of disappeared targets, the authors should also test the relation of the same neural measure with the duration of PFI – or remove this sentence from the Discussion.

Reviewer #1:

In this paper, Davidson et al. characterize the neural correlates of visual disappearance during perceptual filling-in (PFI) using steady-state visual evoked potentials (SSVEPs). They show that target disappearance actually leads to an increase rather than to a decrease of the target SNR. This finding is potentially of importance and the study might ultimately lead to a publication in *eLife*. However, the current version of the manuscript does not provide enough details regarding the underlying assumptions and neural mechanisms. The results should also be better described, interpreted and compared to the existing literature. I list my most substantive concerns below.

1) I was a bit frustrated to see that almost no discussion about the neural mechanisms underlying the results is provided. It seems important to better explain the cortical processes involved (e.g. the authors could compare more carefully their results with those obtained in macaque electrophysiology by De Weerd et al., 1995).To go further along this direction, one possibility would also be to analyse the SNRs at the intermodulation frequencies (I see in Figure 2—figure supplement 2 that responses at F2-F1 = 5Hz are significantly above noise). This would permit to characterize and discuss the interactions between the neural responses corresponding to the processing of the targets and to the surround (see e.g. Appelbaum et al., 2008).

2) When I read the whole manuscript, I had the feeling that the analysis of the SNR change latencies (which is currently described in the supplements) would deserve to be more documented and to appear in the main document. The finding that changes in background SNR precede changes in target SNR is an important result which clarifies the temporal sequence of neural activations. That would also be nice if the authors could determine when the SNR change corresponding to the inter-modulation product (e.g. at F2-F1) appears (see my first point above).

3) To better characterize the difference between the responses to PFI vs. to phenomenally matched disappearances (PMD) and support the claim that target-SNR decreases rather than increases during PMD (subsection “Increased SNR during invisibility is unique to targets during PFI”), that would be great to show the target-SNR changes around button press (i.e. the equivalent of Figure 2B and E) for PMD.

4) The target disappearance during PFI is associated with an increase of SNR and therefore, SSVEPs in this case do not reflect conscious perception. But does it necessarily imply that this target-SNR increase reflects attention instead? The authors base their interpretation on previous studies (Lou, 1999; De Weerd et al., 2006) where attending to target feature increased PFI probability (which I think is not exactly equivalent to the PFI magnitude reported here) and also on the correlation they found between target-SNR and evoked alpha. However, these are indirect evidences and in their experimental protocol, attention was not directly manipulated (as e.g. in Morgan et al., 1996 or Müller et al., 2006). I would suggest being a little bit more cautious with this interpretation in the manuscript.

5) Before this study, other groups looked at the dissociation between attention and perceptual awareness (among others, see e.g. Wyart and Tallon-Baudry, 2008; 2009; Koivisto et al., 2009; Norman et al., 2013). A deeper review of the existing literature on this topic (in the Introduction and/or Discussion) would permit to better understand what is already known and also to provide leads for future investigations.

Reviewer #2:

Overall, I think this is a creative study, with very interesting findings. A major weakness is that the interpretations seem a bit exaggerated and alternative interpretations not considered. Using a creative paradigm of perceptual filling-in, the authors show that increased attention (indexed by a reduction in alpha power over central-parietal locations, and supported by previous psychophysics studies) is associated with perceptual filling-in, and the phenomenal disappearance of targets. By tagging targets and surround with different frequencies, they show that SSVEP elicited by targets increases at the time of perceptual filling-in. These results suggest that SSVEP, thought to index the content of visual perception in previous binocular rivalry studies, can be dissociated from conscious perception in this paradigm, and instead reflect attention.

While the results are interesting and novel, they are perhaps not as surprising as the authors present them to be. Given that previous studies have shown a clear connection between SSVEP and attention (e.g. Morgan, Hansen and Hillyard, 1996, cited by the authors), these results shown that when attention and awareness are dissociated (as the last author has nicely demonstrated/argued previously), SSVEP goes with attention.

These results do not demonstrate that all sensory-cortical activity goes along with attention instead of awareness, as the authors' Abstract/significance statement/Discussion suggest to be the case. E.g., in Abstract/significance statement, the authors only state "neural activity" or "neural response", instead of specifically SSVEP, which can be misleading. Similarly, in the Discussion, it remains a possibility that other types of neural activity (e.g. spiking rate or recurrent activity) in sensory cortex correlates with the vividness of conscious experience, which would in principle be consistent with first-order or GNW theories.

An analysis comment:

In the Discussion, the authors mention "As more targets disappeared and presumably drew attention, both the duration of their absence and strength of target SNR increased." The duration effect, shown in supplementary materials, is not referenced in the main text as I as I could find. In Figure 2, in addition to investigating SSVEP's relation with the number of disappeared targets, the authors could also test its relation with the duration of PFI.

---

## [Author Response]

Essential revisions:1) Novelty of the findings. Although the findings are interesting and novel in this particular context of perceptual filling-in, they may not be as surprising as the authors present them to be. Indeed, previous work (including some cited by the authors, e.g., Morgan, Hansen and Hillyard, 1996) has shown a clear connection between SSVEP and attention. And it is also known from previous work that PFI increases with attention (Lou, 1999; De Weerd, Smith and Greenberg, 1996). The present finding adds to this existing body of work by showing that when PFI (a "visual disappearance" effect) is used to dissociate awareness and attention, SSVEP follows attention rather than consciousness. In other words, rather than showing that SSVEP is a general neural measure of invisibility, an alternative interpretation of the finding is that SSVEP is a neural measure of attention, which in this particular setting of frequency-tagged PFI becomes a neural measure of invisibility (see also the second concern below).

We agree with the reviewers here, and do not claim that the SSVEP is a general neural measure of invisibility. Instead, we argue that SSVEP strength is a neural measure of attention.

To clarify, we have amended the title of our paper into:

“The SSVEP tracks attention, not consciousness, during perceptual filling-in”

We have also removed statements that could be mistaken as claims the SSVEP is a general measure of invisibility.

As you see below, we have also extended our analysis and Discussion to buttress our argument that the SSVEP is more likely to be a neural measure of attention than to capture the current contents of visual phenomenology.

With regard to the novelty of our findings, we believe that this latter distinction is important, and apologise that this had not been made clearer. Indeed, we believe that one of the novel contributions which is afforded by our unique experimental design is the clarification of the relationship between SSVEPs, attention and visual conscious perception. It is true that previous work has linked SSVEP strength to attention. Crucially however, a parallel literature has existed which assumes SSVEP strength also captures visual phenomenology.

For example, one influential paper (Zhang et al., 2011), used the strength of SSVEPs as a proxy for conscious contents during binocular rivalry. This has been a foundational paper to argue that binocular rivalry requires visual attention due to their key manipulation – a comparison of SSVEP modulation between attended and unattended conditions. The authors found the depth of SSVEP modulation to be reduced during unattended rivalry, and argued as a result that binocular rivalry had ceased. We can now revise these claims to state that rivalry may well have continued, as SSVEPs are a neural measure related to attentional amplification and suppression, and not phenomenal vision. This distinction was not possible before dissociating attention and awareness as we have done in our paradigm.

This is just one example, and many older (e.g. Lansing, 1964; Tononi et al., 1998) and new (e.g. Bock et al., 2019) experiments are similarly based on the premise that SSVEPs correlate with visual phenomenology without considering the influence of attention.

We have now highlighted this novel contribution in our Discussion, by summarising the above example for the audience of *eLife*:

“In one notable study, a reduction in SSVEP strength was interpreted to indicate the cessation of binocular rivalry entirely (Zhang et al., 2011). […] Another viable, and potentially simpler explanation, is that SSVEP strength instead reflects attention, which is typically positively correlated with conscious perception”.

2) The Abstract and the Discussion should be revised in a way that considers the finding beyond the specific context of PFI used in this study. For example, the suggestion that the "brain actively represents disappearing, or absent stimuli" should be accompanied by the alternative interpretation that SSVEP reflects a neural measure of attention rather than awareness, which in the particular setting of PFI go in opposite directions. Also, as raised by reviewer #1, previous work has studied the dissociation between the neural correlates of visual attention and awareness (among others, see e.g. Wyart and Tallon-Baudry, 2008; 2009; Koivisto et al., 2009; Norman et al., 2013). A deeper review of the existing literature on this topic (in the Introduction and/or Discussion) would afford to better understand what is already known and also to provide insights for future work.

Thank you for these suggestions to expand the context of our work and consider alternate interpretations. We have now rewritten our Abstract, Introduction, and Discussion to accommodate alternative explanations, and to deepen our review of existing literature on this topic.

Abstract:

“Research on the neural basis of conscious perception has almost exclusively shown that becoming aware of a stimulus leads to increased neural responses. […] Instead we conclude that SSVEP strength more closely measures changes in attention.”

Our Introduction has also been revised to consider the broader context of attention and consciousness research, beyond the example of PFI, used in the present study:

For example, from our Introduction:

“In the behavioural and neuronal study of consciousness and attention, PFI has an unusual and potentially very revealing property: directing attention toward targets in the visual periphery facilitates perceptual disappearance (De Weerd, Smith, and Greenberg, 2006; Lou, 1999). […] Here, we demonstrate that PFI indeed dissociates neural measures which normally positively correlate with both attention to – and the conscious perception of, visual stimuli.”

In our Discussion, we have also now included a new subsection, that positions our results more generally in the context of what is known about visual attention and awareness. We also discuss alternative explanations, and provide recommendations for future work. See Discussion subsection “Visual attention and conscious perception”

3) Nature of the relation between SSVEP/alpha power and attention. The authors ground their interpretation on previous studies (Lou, 1999; De Weerd et al., 2006) where attending to target features increased PFI probability, and also on the correlation they found between target-tagged SNR and alpha power. However, these are both indirect evidence – in the case of alpha power, it amounts to reverse inference to assign changes in alpha power to changes in attention. And attention was not directly manipulated in the present study, which would have provided a much stronger monitoring of the relation between attention and awareness (see, e.g., Morgan et al., 1996 or Müller et al., 2006). We thus suggest the authors to be more cautious regarding the interpretations that they seek to draw regarding attention in their study. Attention was not manipulated experimentally, and using alpha power as a measure of attention amounts to reverse inference.

We agree with this point, and have now explicitly acknowledged the indirect nature of this evidence in our Discussion:

“Importantly, our interpretation that SSVEPs reflect attention but not consciousness is supported by indirect, yet convergent evidence”.

We agree that a direct manipulation of attention would be a strong test of our claims, and specify this as a possibility for future work:

“To tease apart the neural correlates of conscious phenomenology from those of attention and reports, future work may be able to test whether SSVEPs during PFI at attended / unattended locations replicates the pattern which we have described, …”

However, we also maintain that an explanation based on attention is the most parsimonious interpretation of our results.

To justify our claims, we have now included an extended discussion of the convergent evidence, as well as evidence against alternative explanations based on report-related motor selection, somatosensory spatial attention, arousal, and vigilance.

We also note that our interpretation based on attention has been strengthened by extending our analysis of PFI and PMD SNR timing differences (see reviewer comment 7, below).

4) Specificity of the finding. These results presented in the manuscript do not demonstrate that all sensory-cortical activity goes along with attention instead of awareness, as the authors' Abstract/significance statement/Discussion suggest to be the case ("by showing that the strength of stimulus-specific neural activity can encode the conscious absence of a stimulus", or "we found that neural activity increased […] when targets became invisible"). In Abstract and significance statement, the authors only state "neural activity" or "neural response", instead of SSVEP, which can be misleading. Similarly, in the Discussion, the authors should note that it remains entirely possible that other types of neural activity (e.g. spike rate or recurrent activity) in visual cortex correlates with the vividness of conscious experience, which would in principle be consistent with existing theories of visual awareness (e.g., GNW).

Thank you for this suggestion, we did not mean to imply that our finding will generalize to all sensory-cortical activities. We have now amended our Abstract, Introduction, and Discussion to focus on the specificity of the effect regarding SSVEPs.

In our Abstract:

“We show that in a perceptual filling-in paradigm the disappearance of a stimulus and subjective invisibility are associated with increases in neural activity, as measured with steady-state visually evoked potentials (SSVEP), in electroencephalography (EEG). […] These findings cast doubt on the direct relationship previously reported between the strength of neural activity and conscious perception, at least when measured with current tools, such as the SSVEP.”

We have also now included a discussion of the likelihood that other types of activity (spike-rates) correlate with conscious experience, and that the link between spike-rates and SSVEPs are not well established:

“Insofar as the strength of target SSVEP signals when measured with EEG may reflect the strength of target-specific spike activity in the sensory cortex, our finding does challenge such theories that propose response strength is positively correlated with the vividness of conscious experience. Currently, however, the exact mechanisms of SSVEP generation are not well understood (Kawashima et al., 2020; Kim et al., 2017; Norcia et al., 2015; Notbohm, Kurths and Herrmann, 2016; Rager and Singer, 1998; Salelkar and Ray, 2020), and future studies will be necessary to clarify their input.”

We also discuss previous literature regarding the link between spike-rates in visual cortex and conscious experience. We also paraphrase the important point made above, that the patterns of activity, which includes recurrent processing may correlate with phenomenology:

“Meanwhile, there exists another family of theories that propose that certain types of causal interactions underlie conscious phenomenology, such as recurrent processing theory (Lamme, 2006, 2014; Lamme and Roelfsema, 2000) and integrated information theory (Haun and Tononi, 2019; Tononi, 2004). For these theories, it is not the strength, but the patterns of neuronal connectivity and their states that determine phenomenology as a whole.”

5) Comparison between PFI and phenomenally matched disappearances (PMD). To better characterize the difference between the responses to PFI vs. to PMD, and support the claim that target-tagged SNR decreases rather than increases during PMD (subsection “Increased SNR during invisibility is unique to targets during PFI”), it would be very important to show the target-tagged SNR changes around button press for PMD – i.e. the equivalent of Figure 2B and E for PMD.

We have now revised Figure 2 to include the SNR time-course during PMD.

6) Neural mechanisms underlying the observed findings. Almost no discussion about the neural mechanisms underlying the surprising finding is provided. It seems important to better explain the cortical processes involved (e.g., the authors could compare more carefully their results with those obtained in macaque electrophysiology by De Weerd et al., 1995). At the analytical level, one possibility would also be to analyse the SSVEP SNR at the intermodulation frequencies (I see in Figure 2—figure supplement 2 that responses at F2 minus F1 = 5Hz are significantly above noise). This would permit to characterize and discuss the interactions between the neural responses corresponding to the processing of the targets and to those corresponding to the processing of the surround (see e.g. Appelbaum et al., 2008).

We agree with this suggestion, and have now incorporated a discussion of the neural mechanisms of PFI. Both in the main text, where we have moved previous supplementary analysis on SNR change latencies, and in our Discussion.

In addition, we have now extended our analysis of SNR change latencies (discussed below), to include the IM, as well as a comparison between PFI and PMD periods. This extended analysis offers new insights in favour of an active model of target-surround interaction during PFI. See our Results subsection “Changes in surround-SNR precede target-SNR during PFI”

7) Analysis of SNR change latencies. The analysis of SNR change latencies (currently described in the supplements) would deserve to be more documented and to appear in the main document. The finding that changes in background SNR precede changes in target SNR is an important result which clarifies the temporal sequence of neural activations. It would be interesting to determine when the SNR change corresponding to the inter-modulation product (e.g., at F2 minus F1) appears (see the previous concern).

Thank you for this suggestion. We have now moved our analysis of SNR change latencies into the main document, and expanded this analysis to include changes in the SNR of the IM component. We have also extended our analysis to PMD periods, in order to distinguish the neural mechanisms during PFI from PMD. New Figure 4.

As well as an extended treatment of the neural mechanisms of PFI in the main text, we now position our results with relevance to previous PFI research in our Discussion, with a new subsection “Neural mechanisms of perceptual filling-in”.

8) Relation between duration of PFI and SSVEP. In the Discussion, the authors mention: "As more targets disappeared and presumably drew attention, both the duration of their absence and strength of target SNR increased." However, the duration effect, shown in supplementary materials, is not referenced in the main text. In Figure 2, in addition to investigating the relation of SSVEP with the number of disappeared targets, the authors should also test the relation of the same neural measure with the duration of PFI – or remove this sentence from the Discussion.

Thank you for the opportunity to clarify our analysis. We have now more explicitly referenced the duration effect in the main text:

“We first confirmed in our behavioural analysis that targets disappeared together more frequently than in isolation (see subsection “A spatial interaction between target locations increases PFI duration when targets disappear together”; Figure 2—figure supplement 1). Consistent with our prior work, we also replicate that the duration of PFI increases with an increasing number of invisible targets, strongly suggesting a facilitatory grouping mechanism across visual hemifields and target locations (Davidson et al., 2020).”

As for the suggestion to extend our SSVEP analysis to PFI duration, we wish to clarify that the ‘amount’ of PFI is not just the number of absent targets, but also includes duration information. We quantified the ‘amount’ of PFI based on our event-by-event image analysis, which is the duration-weighted number of buttons pressed over a 3 second period.

To clarify this in our main text we have now revised the wording throughout our main text and Materials and methods e.g.:

“…measuring the “amount of PFI”, which we defined as the duration-weighted number of invisible targets (see Materials and methods)”

Reviewer #1:In this paper, Davidson et al. characterize the neural correlates of visual disappearance during perceptual filling-in (PFI) using steady-state visual evoked potentials (SSVEPs). They show that target disappearance actually leads to an increase rather than to a decrease of the target SNR. This finding is potentially of importance and the study might ultimately lead to a publication in eLife. However, the current version of the manuscript does not provide enough details regarding the underlying assumptions and neural mechanisms. The results should also be better described, interpreted and compared to the existing literature. I list my most substantive concerns below.

Thank you for your detailed review. We have provided the details that you requested, and improved on our manuscript by extending our literature review. Some of the points that we have already answered above are not repeated here.

1) I was a bit frustrated to see that almost no discussion about the neural mechanisms underlying the results is provided. It seems important to better explain the cortical processes involved (e.g. the authors could compare more carefully their results with those obtained in macaque electrophysiology by De Weerd et al., 1995).To go further along this direction, one possibility would also be to analyse the SNRs at the intermodulation frequencies (I see in Figure 2—figure supplement 2 that responses at F2-F1 = 5Hz are significantly above noise). This would permit to characterize and discuss the interactions between the neural responses corresponding to the processing of the targets and to the surround (see e.g. Appelbaum et al., 2008).

See above (comment 6-7).

2) When I read the whole manuscript, I had the feeling that the analysis of the SNR change latencies (which is currently described in the supplements) would deserve to be more documented and to appear in the main document. The finding that changes in background SNR precede changes in target SNR is an important result which clarifies the temporal sequence of neural activations. That would also be nice if the authors could determine when the SNR change corresponding to the inter-modulation product (e.g. at F2-F1) appears (see my first point above).

Thank you for this idea. We have now included the analysis of IM SNR in our treatment of PFI mechanisms. This extension is expanded on above, (comment 6-7).

3) To better characterize the difference between the responses to PFI vs. to phenomenally matched disappearances (PMD) and support the claim that target-SNR decreases rather than increases during PMD (subsection “Increased SNR during invisibility is unique to targets during PFI”), that would be great to show the target-SNR changes around button press (i.e. the equivalent of Figure 2B and E) for PMD.

See above (comment 5).

4) The target disappearance during PFI is associated with an increase of SNR and therefore, SSVEPs in this case do not reflect conscious perception. But does it necessarily imply that this target-SNR increase reflects attention instead? The authors base their interpretation on previous studies (Lou, 1999; De Weerd et al., 2006) where attending to target feature increased PFI probability (which I think is not exactly equivalent to the PFI magnitude reported here) and also on the correlation they found between target-SNR and evoked alpha. However, these are indirect evidences and in their experimental protocol, attention was not directly manipulated (as e.g. in Morgan et al., 1996 or Müller et al., 2006). I would suggest being a little bit more cautious with this interpretation in the manuscript.

See above (comment 4).

5) Before this study, other groups looked at the dissociation between attention and perceptual awareness (among others, see e.g. Wyart and Tallon-Baudry, 2008; 2009; Koivisto et al., 2009; Norman et al., 2013). A deeper review of the existing literature on this topic (in the Introduction and/or Discussion) would permit to better understand what is already known and also to provide leads for future investigations.

See above (comment 2).

Reviewer #2:Overall, I think this is a creative study, with very interesting findings. A major weakness is that the interpretations seem a bit exaggerated and alternative interpretations not considered. Using a creative paradigm of perceptual filling-in, the authors show that increased attention (indexed by a reduction in alpha power over central-parietal locations, and supported by previous psychophysics studies) is associated with perceptual filling-in, and the phenomenal disappearance of targets. By tagging targets and surround with different frequencies, they show that SSVEP elicited by targets increases at the time of perceptual filling-in. These results suggest that SSVEP, thought to index the content of visual perception in previous binocular rivalry studies, can be dissociated from conscious perception in this paradigm, and instead reflect attention.While the results are interesting and novel, they are perhaps not as surprising as the authors present them to be. Given that previous studies have shown a clear connection between SSVEP and attention (e.g. Morgan, Hansen and Hillyard, 1996, cited by the authors), these results shown that when attention and awareness are dissociated (as the last author has nicely demonstrated/argued previously), SSVEP goes with attention.

See above (comment 1).

These results do not demonstrate that all sensory-cortical activity goes along with attention instead of awareness, as the authors' Abstract/significance statement/Discussion suggest to be the case. E.g., in Abstract/significance statement, the authors only state "neural activity" or "neural response", instead of specifically SSVEP, which can be misleading. Similarly, in the Discussion, it remains a possibility that other types of neural activity (e.g. spiking rate or recurrent activity) in sensory cortex correlates with the vividness of conscious experience, which would in principle be consistent with first-order or GNW theories.

See above (comment 4).

An analysis comment:In the Discussion, the authors mention "As more targets disappeared and presumably drew attention, both the duration of their absence and strength of target SNR increased." The duration effect, shown in supplementary materials, is not referenced in the main text as I could find. In Figure 2, in addition to investigating SSVEP's relation with the number of disappeared targets, the authors could also test its relation with the duration of PFI.

See above (comment 8).